# RAPTOR: RECURSIVE ABSTRACTIVE PROCESSING FOR TREE-ORGANIZED RETRIEVAL

**Parth Sarthi, Salman Abdullah, Aditi Tuli, Shubh Khanna, Anna Goldie, Christopher D. Manning**
Stanford University
psarthi@cs.stanford.edu

## ABSTRACT

Retrieval-augmented language models can better adapt to changes in world state and incorporate long-tail knowledge. However, most existing methods retrieve only short contiguous chunks from a retrieval corpus, limiting holistic understanding of the overall document context. We introduce the novel approach of recursively embedding, clustering, and summarizing chunks of text, constructing a tree with differing levels of summarization from the bottom up. At inference time, our RAPTOR model retrieves from this tree, integrating information across lengthy documents at different levels of abstraction. Controlled experiments show that retrieval with recursive summaries offers significant improvements over traditional retrieval-augmented LMs on several tasks. On question-answering tasks that involve complex, multi-step reasoning, we show state-of-the-art results; for example, by coupling RAPTOR retrieval with the use of GPT-4, we can improve the best performance on the QuALITY benchmark by 20% in absolute accuracy.

## 1 INTRODUCTION

Large Language Models (LLMs) have emerged as transformative tools showing impressive performance on many tasks. With the growing size of LLMs, they can serve standalone as very effective knowledge stores, with facts encoded within their parameters (Petroni et al., 2019; Jiang et al., 2020; Talmor et al., 2020; Rae et al., 2021; Hoffmann et al., 2022; Chowdhery et al., 2022; Bubeck et al., 2023; Kandpal et al., 2023) and models can be further improved with fine-tuning on downstream tasks (Roberts et al., 2020). Nevertheless, even a large model does not contain sufficient domain-specific knowledge for particular tasks and the world continues to change, invalidating facts in the LLM. Updating the knowledge of these models through additional fine-tuning or editing is difficult, particularly when dealing with vast text corpora (Lewis et al., 2020; Mitchell et al., 2022). An alternative approach, pioneered in open domain question answering systems (Chen et al., 2017; Yu et al., 2018), is to index large quantities of text, after splitting it into chunks (paragraphs), in a separate information retrieval system. Retrieved information is then presented to the LLM along with the question as context ("retrieval augmentation", Lewis et al., 2020; Izacard et al., 2022; Min et al., 2023; Ram et al., 2023), making it easy to provide a system with current knowledge particular to some domain and enabling easy interpretability and provenance tracking, whereas the parametric knowledge of LLMs is opaque and difficult to trace back to its source (Akyurek et al., 2022).

Nevertheless, existing retrieval-augmented approaches also have flaws. The one we tackle is that most existing methods retrieve only a few short, contiguous text chunks, which limits their ability to represent and leverage large-scale discourse structure. This is particularly relevant for thematic questions that require integrating knowledge from multiple parts of a text, such as understanding an entire book, as in the NarrativeQA dataset (Kočiskỳ et al., 2018). Consider the fairy tale of Cinderella, and the question "How did Cinderella reach her happy ending?". The top-$k$ retrieved short contiguous texts will not contain enough context to answer the question.

To address this, we design an indexing and retrieval system that uses a tree structure to capture both high-level and low-level details about a text. As shown in Figure 1, our system, RAPTOR, clusters chunks of text, generates text summaries of those clusters, and then repeats, generating a tree from the bottom up. This structure enables RAPTOR to load into an LLM's context chunks representing the text at different levels so that it can effectively and efficiently answer questions at different levels.

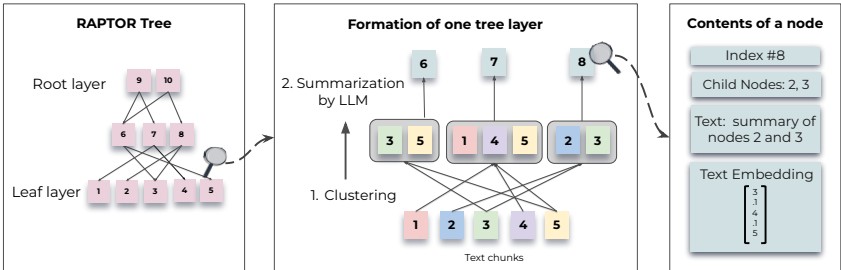

Figure 1: **Tree construction process:** RAPTOR recursively clusters chunks of text based on their vector embeddings and generates text summaries of those clusters, constructing a tree from the bottom up. Nodes clustered together are siblings; a parent node contains the text summary of that cluster.

Our main contribution is the idea of using text summarization to allow retrieval augmentation of context at different scales, and to show its effectiveness in experiments on collections of long documents. Controlled experiments with three language models (UnifiedQA (Khashabi et al., 2020), GPT-3 (Brown et al., 2020) and GPT-4 (OpenAI, 2023)) show that RAPTOR outperforms current retrieval augmentation. Moreover, RAPTOR coupled with GPT-4, and sometimes even with UnifiedQA, gives new state-of-the-art results on three QA tasks: free text response questions on books and movies (NarrativeQA, Kočiskỳ et al. 2018), full-text NLP papers (QASPER, Dasigi et al. 2021), and multiple-choice questions based on medium-length passages (QuALITY, Pang et al. 2022).[1]

## 2 RELATED WORK

**Why Retrieval?** Recent advances in hardware and algorithms have indeed expanded the context lengths that models can handle, leading to questions about the need for retrieval systems (Dai et al., 2019; Dao et al., 2022; Liu et al., 2023). However, as Liu et al. (2023) and Sun et al. (2021) have noted, models tend to underutilize long-range context and see diminishing performance as context length increases, especially when pertinent information is embedded within a lengthy context. Moreover, practically, use of long contexts is expensive and slow. This suggests that selecting the most relevant information for knowledge-intensive tasks is still crucial.

**Retrieval Methods** Retrieval-augmented language models (RALMs) have seen improvements in various components: the retriever, the reader, and end-to-end system training. Retrieval methods have transitioned from traditional term-based techniques like **TF-IDF** (Spärck Jones, 1972) and **BM25** (Robertson et al., 1995; Roberts et al., 2020) to deep learning–based strategies (Karpukhin et al., 2020; Khattab & Zaharia, 2020; Sachan et al., 2023). Some recent work proposes using large language models as retrievers due to their ability to memorize extensive knowledge (Yu et al., 2022; Sun et al., 2022). Research on the reader component includes **Fusion-in-Decoder (FiD)** (Izacard & Grave, 2022), which employs both DPR and BM25 for retrieval and processes passages independently in the encoder and **RETRO** (Borgeaud et al., 2022; Wang et al., 2023), which utilizes cross-chunked attention and chunkwise retrieval to generate text grounded on retrieved context.

End-to-end system training work includes **Atlas** (Izacard et al., 2022), which fine-tunes an encoder-decoder model in conjunction with the retriever; **REALM** (Guu et al., 2020), a bidirectional, masked LM fine-tuned for open-domain question answering; and **RAG (Retrieval-Augmented Generation)** (Lewis et al., 2020), which integrates pre-trained sequence-to-sequence models with a neural retriever. Min et al. (2021) introduced **Joint Passage Retrieval (JPR)** model which uses a tree-decoding algorithm to handle passage diversity and relevance in multi-answer retrieval. **Dense Hierarchical Retrieval (DHR)** and **Hybrid Hierarchical Retrieval (HHR)** represent advancements in retrieval accuracy by combining document and passage level retrievals and integrating sparse and dense retrieval methods, respectively (Liu et al., 2021; Arivazhagan et al., 2023).

---

[1]We have released the code of RAPTOR at https://github.com/parthsarthi03/raptor.

Despite a diversity in methods, the retrieving components of models predominantly rely on standard approaches, i.e., chunking corpora and encoding with BERT-based retrievers. Although this approach is widely adopted, Nair et al. (2023) highlights a potential shortcoming: contiguous segmentation might not capture the complete semantic depth of the text. Reading extracted snippets from technical or scientific documents may lack important context making them difficult to read or even misleading. (Cohan & Goharian, 2017; Newman et al., 2023; Zhang et al., 2023).

**Recursive summarization as Context**   Summarization techniques provide a condensed view of documents, enabling more focused engagement with the content (Angelidis & Lapata, 2018). The summarization/snippet model by Gao et al. (2023) uses summarizations and snippets of passages, which improves correctness on most datasets but can sometimes be a lossy means of compression. The recursive-abstractive summarization model by Wu et al. (2021) employs task decomposition to summarize smaller text chunks, which are later integrated to form summaries of larger sections. While this method is effective for capturing broader themes, it can miss granular details. LlamaIndex (Liu, 2022) mitigates this issue by similarly summarizing adjacent text chunks but also retaining intermediate nodes thus storing varying levels of detail, keeping granular details. However, both methods, due to their reliance on adjacency for grouping or summarizing adjacent nodes, may still overlook distant interdependencies within the text, which we can find and group with RAPTOR.

## 3   METHODS

**Overview of RAPTOR**   Building on the idea that long texts often present subtopics and hierarchical structures (Cao & Wang, 2022; Dong et al., 2023b), RAPTOR addresses the issue of semantic depth and connection in reading by building a recursive tree structure that balances broader thematic comprehension with granular details and which allows nodes to be grouped based on semantic similarity not just order in the text.

Construction of the RAPTOR tree begins with segmenting the retrieval corpus into short, contiguous texts of length 100, similar to traditional retrieval augmentation techniques. If a sentence exceeds the 100-token limit, we move the entire sentence to the next chunk, rather than cutting it mid-sentence. This preserves the contextual and semantic coherence of the text within each chunk. These texts are then embedded using SBERT, a BERT-based encoder (`multi-qa-mpnet-base-cos-v1`) (Reimers & Gurevych, 2019). The chunks and their corresponding SBERT embeddings form the leaf nodes of our tree structure.

To group similar text chunks, we employ a clustering algorithm. Once clustered, a Language Model is used to summarize the grouped texts. These summarized texts are then re-embedded, and the cycle of embedding, clustering, and summarization continues until further clustering becomes infeasible, resulting in a structured, multi-layered tree representation of the original documents. An important aspect of RAPTOR is its computational efficiency. The system scales linearly in terms of both build time and token expenditure, making it suitable for processing large and complex corpora. For a comprehensive discussion on RAPTOR's scalability, please refer to the Appendix A.

For querying within this tree, we introduce two distinct strategies: tree traversal and collapsed tree. The tree traversal method traverses the tree layer-by-layer, pruning and selecting the most relevant nodes at each level. The collapsed tree method evaluates nodes collectively across all layers to find the most relevant ones.

**Clustering Algorithm**   Clustering plays a key role in building the RAPTOR tree, organizing text segments into cohesive groups. This step groups related content together, which helps the subsequent retrieval process.

One of the unique aspects of our clustering approach is the use of soft clustering, where nodes can belong to multiple clusters without requiring a fixed number of clusters. This flexibility is essential because individual text segments often contain information relevant to various topics, thereby warranting their inclusion in multiple summaries.

Our clustering algorithm is based on Gaussian Mixture Models (GMMs), an approach that offers both flexibility and a probabilistic framework. GMMs assume that data points are generated from a mixture of several Gaussian distributions.

Given a set of $N$ text segments, each represented as a $d$-dimensional dense vector embedding, the likelihood of a text vector, $\mathbf{x}$, given its membership in the $k^{th}$ Gaussian distribution, is denoted by $P(\mathbf{x}|k) = \mathcal{N}(\mathbf{x}; \mu_k, \mathbf{\Sigma}_k)$. The overall probability distribution is a weighted combination $P(\mathbf{x}) = \sum_{k=1}^{K} \pi_k \mathcal{N}(\mathbf{x}; \mu_k, \mathbf{\Sigma}_k)$, where $\pi_k$ signifies the mixture weight for the $k^{\text{th}}$ Gaussian distribution.

The high dimensionality of vector embeddings presents a challenge for traditional GMMs, as distance metrics may behave poorly when used to measure similarity in high-dimensional spaces (Aggarwal et al., 2001). To mitigate this, we employ Uniform Manifold Approximation and Projection (UMAP), a manifold learning technique for dimensionality reduction (McInnes et al., 2018). The number of nearest neighbors parameter, $n\_neighbors$, in UMAP determines the balance between the preservation of local and global structures. Our algorithm varies $n\_neighbors$ to create a hierarchical clustering structure: it first identifies global clusters and then performs local clustering within these global clusters. This two-step clustering process captures a broad spectrum of relationships among the text data, from broad themes to specific details.

Should a local cluster's combined context ever exceed the summarization model's token threshold, our algorithm recursively applies clustering within the cluster, ensuring that the context remains within the token threshold.

To determine the optimal number of clusters, we employ the Bayesian Information Criterion (BIC) for model selection. BIC not only penalizes model complexity but also rewards goodness of fit (Schwarz, 1978). The BIC for a given GMM is $BIC = \ln(N)k - 2\ln(\hat{L})$, where $N$ is the number of text segments (or data points), $k$ is the number of model parameters, and $\hat{L}$ is the maximized value of the likelihood function of the model. In the context of GMM, the number of parameters $k$ is a function of the dimensionality of the input vectors and the number of clusters.

With the optimal number of clusters determined by BIC, the Expectation-Maximization algorithm is then used to estimate the GMM parameters, namely the means, covariances, and mixture weights.

While the Gaussian assumption in GMMs may not perfectly align with the nature of text data, which often exhibits a sparse and skewed distribution, our empirical observations suggest that it offers an effective model for our purpose. We run an ablation comparing GMM Clustering with summarizing contiguous chunks and provide details in Appendix B.

**Model-Based Summarization**   After clustering the nodes using Gaussian Mixture Models, the nodes in each cluster are sent to a language model for summarization. This step allows the model to transform large chunks of text into concise, coherent summaries of the selected nodes. For our experiments, we use gpt-3.5-turbo to generate the summaries. The summarization step condenses the potentially large volume of retrieved information into a manageable size. We provide statistics on the compression due to the summarization in Appendix C and the prompt used for summarization in Appendix D.

While the summarization model generally produces reliable summaries, a focused annotation study revealed that about 4% of the summaries contained minor hallucinations. These did not propagate to parent nodes and had no discernible impact on question-answering tasks. For an in-depth analysis of hallucinations, refer to the appendix E.

**Querying**   In this section, we elaborate on the two querying mechanisms employed by RAPTOR: tree traversal and collapsed tree. These methods offer unique ways of traversing the multi-layered RAPTOR tree to retrieve relevant information, each with its own advantages and trade-offs. We provide the pseudocode of both methods in Appendix F. Note that we embed all nodes using SBERT.

The **tree traversal** method first selects the top-k most relevant root nodes based on their cosine similarity to the query embedding. The children of these selected nodes are considered at the next layer and the top-k nodes are selected from this pool again based on their cosine similarity to the query vector. This process is repeated until we reach the leaf nodes. Finally, the text from all selected nodes is concatenated to form the retrieved context. The algorithm's steps are outlined below:

1. Start at the root layer of the RAPTOR tree. Compute the cosine similarity between the query embedding and the embeddings of all nodes present at this initial layer.

2. Choose the top-$k$ nodes based on the highest cosine similarity scores, forming the set $S_1$.

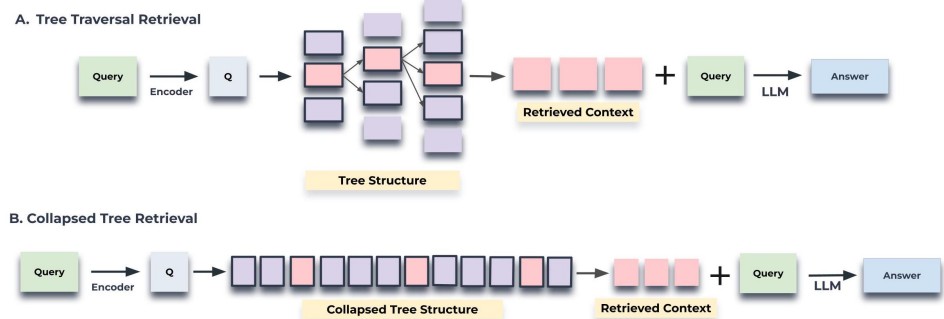

Figure 2: **Illustration of the tree traversal and collapsed tree retrieval mechanisms.** Tree traversal starts at the root level of the tree and retrieves the top-$k$ (here, top-1) node(s) based on cosine similarity to the query vector. At each level, it retrieves the top-$k$ node(s) from the child nodes of the previous layer's top-$k$. Collapsed tree collapses the tree into a single layer and retrieves nodes until a threshold number of tokens is reached, based on cosine similarity to the query vector. The nodes on which cosine similarity search is performed are highlighted in both illustrations.

3. Proceed to the child nodes of the elements in set $S_1$. Compute the cosine similarity between the query vector and the vector embeddings of these child nodes.

4. Select the top $k$ child nodes with the highest cosine similarity scores to the query, forming the set $S_2$.

5. Continue this process recursively for $d$ layers, producing sets $S_1, S_2, \ldots, S_d$.

6. Concatenate sets $S_1$ through $S_d$ to assemble the relevant context to the query.

By adjusting the depth $d$ and the number of nodes $k$ selected at each layer, the tree traversal method offers control over the specificity and breadth of the information retrieved. The algorithm starts with a broad outlook by considering the top layers of the tree and progressively focuses on finer details as it descends through the lower layers.

The **collapsed tree** approach offers a simpler way to search for relevant information by considering all nodes in the tree simultaneously, as depicted in Figure 2. Instead of going layer-by-layer, this method flattens the multi-layered tree into a single layer, essentially bringing all the nodes onto the same level for comparison. The steps for this method are outlined below:

1. First, collapse the entire RAPTOR tree into a single layer. This new set of nodes, denoted as $C$, contains nodes from every layer of the original tree.

2. Next, calculate the cosine similarity between the query embedding and the embeddings of all nodes present in the collapsed set $C$.

3. Finally, pick the top-$k$ nodes that have the highest cosine similarity scores with the query. Keep adding nodes to the result set until you reach a predefined maximum number of tokens, ensuring you don't exceed the model's input limitations.

We tested both approaches on 20 stories from the QASPER dataset. Figure 3 shows the performance of tree traversal with different top- sizes and collapsed tree with different maximum token numbers. The collapsed tree approach consistently performs better. We believe collapsed tree retrieval is better due to offering greater flexibility than tree traversal; i.e., by searching through all the nodes simultaneously, it retrieves information that is at the correct level of granularity for a given question. In comparison, while using tree traversal with the same values of $d$ and $k$, the ratio of nodes from each level of the tree will be constant. So, the ratio of higher-order thematic information to granular details will remain the same regardless of the question.

One drawback, however, of the collapsed tree approach is that it requires cosine similarity search to be performed on all nodes in the tree. However, this can be made more efficient with fast $k$-nearest neighbor libraries such as FAISS (Johnson et al., 2019).

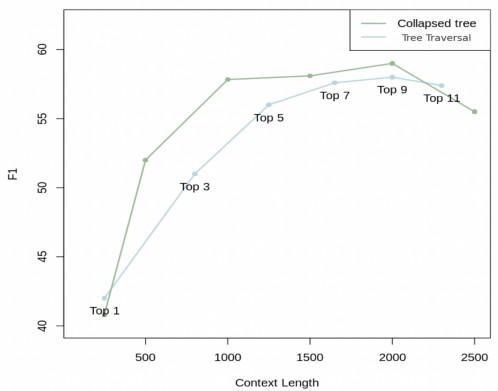

Figure 3: **Comparison of querying methods.** Results on 20 stories from the QASPER dataset using tree traversal with different top-k values, and collapsed tree with different context lengths. Collapsed tree with 2000 tokens produces the best results, so we use this querying strategy for our main results.

Overall, given the collapsed tree approach's greater flexibility and its superior performance on the subset of the QASPER dataset, this is the querying approach with which we proceed. Specifically, we use the collapsed tree with 2000 maximum tokens, which approximately equates to retrieving the top-20 nodes. Using a token-based approach ensures the context does not exceed model context constraints as token counts can vary across nodes. For experiments with the UnifiedQA model, we provide 400 tokens of context, as UnifiedQA has a max context length of 512 tokens. We provide the same amount of tokens of context to RAPTOR and to the baselines.

**Qualitative Study** We conduct a qualitative analysis to understand the benefits of RAPTOR's retrieval process compared to Dense Passage Retrieval (DPR) methods. Our study focuses on thematic, multi-hop questions using a 1500-word Cinderella fairytale. As illustrated in Figure 4, RAPTOR's tree-based retrieval allows it to choose nodes from different tree layers, matching the question's detail level. This approach often yields more relevant and comprehensive information for downstream tasks than DPR. For a detailed discussion and examples, including the text retrieved by both RAPTOR and DPR for specific questions, please refer to the appendix G.

## 4 EXPERIMENTS

**Datasets** We measure RAPTOR's performance across three question-answering datasets: NarrativeQA, QASPER, and QuALITY.

NarrativeQA is a dataset that comprises question-answer pairs based on the full texts of books and movie transcripts, totaling 1,572 documents (Kočiský et al., 2018; Wu et al., 2021). The NarrativeQA-Story task requires a comprehensive understanding of the entire narrative in order to accurately answer its questions, thus testing the model's ability to comprehend longer texts in the literary domain. We measure performance on this dataset using the standard BLEU (B-1, B-4), ROUGE (R-L), and METEOR (M) metrics. Please see appendix H for more details on the NarrativeQA evaluation script used in our experiments.

The QASPER dataset includes 5,049 questions across 1,585 NLP papers, with each question probing for information embedded within the full text (Dasigi et al., 2021). The answer types in QASPER are categorized as Answerable/Unanswerable, Yes/No, Abstractive, and Extractive. Accuracy is measured using standard F1.

Lastly, the QuALITY dataset consists of multiple-choice questions, each accompanied by context passages averaging approximately 5,000 tokens in length (Pang et al., 2022). This dataset calls for reasoning over the entire document for QA tasks, enabling us to measure the performance of our retrieval system on medium-length documents. The dataset includes a challenging subset, QuALITY-HARD, which contains questions that a majority of human annotators answered incorrectly in a speed-setting. We report accuracies for both the entire test set and the HARD subset.

**Controlled Baseline Comparisons** We first present controlled comparisons using the UnifiedQA 3B as the reader, with SBERT (Reimers & Gurevych, 2019), BM25 (Robertson et al., 1995; 2009), and DPR (Karpukhin et al., 2020) as the embedding models with and without the RAPTOR tree structure, on three datasets: QASPER, NarrativeQA, and QuALITY. As shown in Tables 1 and 2,

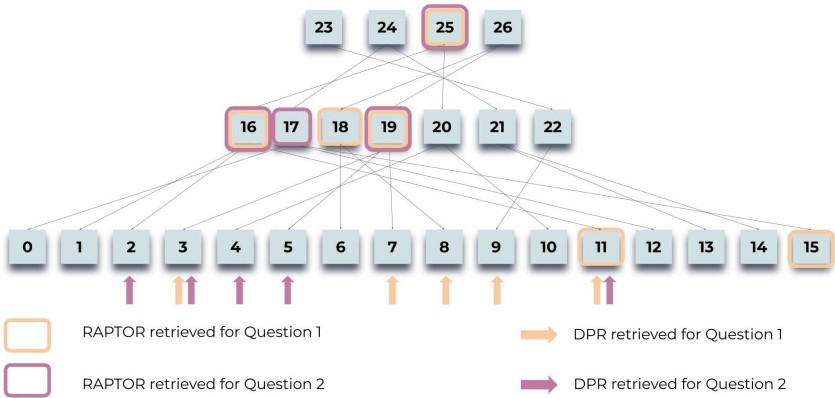

Figure 4: **Querying Process:** Illustration of how RAPTOR retrieves information for two questions about the Cinderella story: "What is the central theme of the story?" and "How did Cinderella find a happy ending?". Highlighted nodes indicate RAPTOR's selections, while arrows point to DPR's leaf nodes. Notably, RAPTOR's context often encompasses the information retrieved by DPR, either directly or within higher-layer summaries.

our results demonstrate that RAPTOR, when combined with any retriever, consistently outperforms the respective retriever across all datasets. [2]

Since RAPTOR with SBERT has the best performance, we use it in all subsequent experiments. We now compare RAPTOR with BM25 and DPR, using three different LLMs: GPT-3, GPT-4, and UnifiedQA. As shown in Table 3, RAPTOR consistently outperforms BM25 and DPR across all three Language Models on the QASPER dataset. RAPTOR's F-1 Match scores are 53.1%, 55.7%, and 36.6% when using GPT-3, GPT-4, and UnifiedQA, respectively. These scores surpass DPR by margins of 1.8, 2.7, and 4.5 points, and outdo BM25 by 6.5, 5.5, and 10.2 points across the respective LLMs. QASPER requires synthesizing information within NLP papers, so it is unsurprising that RAPTOR's higher-level summary nodes would allow it to outperform methods that can only extract the top-$k$ most similar raw chunks of text, which may not contain the correct response in isolation.

Table 1: **NarrativeQA Performance With + Without RAPTOR:** Performance comparison of various retrieval methods (SBERT, BM25, DPR) with and without RAPTOR on the NarrativeQA dataset, using UnifiedQA-3B as the language model. RAPTOR outperforms baselines of each respective retrieval method.

| Model | ROUGE | BLEU-1 | BLEU-4 | METEOR |
|---|---|---|---|---|
| **SBERT with RAPTOR** | **30.87%** | **23.50%** | **6.42%** | **19.20%** |
| SBERT without RAPTOR | 29.26% | 22.56% | 5.95% | 18.15% |
| **BM25 with RAPTOR** | **27.93%** | **21.17%** | **5.70%** | **17.03%** |
| BM25 without RAPTOR | 23.52% | 17.73% | 4.65% | 13.98% |
| **DPR with RAPTOR** | **30.94%** | **23.51%** | **6.45%** | **19.05%** |
| DPR without RAPTOR | 29.56% | 22.84% | 6.12% | 18.44% |

Likewise, in the QuALITY dataset as shown in Table 4, RAPTOR achieves an accuracy of 62.4%, which is a 2% and 5.1% improvement over DPR and BM25. Similar trends are observed when UnifiedQA is employed, with RAPTOR outperforming DPR and BM25 by 2.7% and 6.7%, respectively.

Finally, in the NarrativeQA dataset, as presented in Table 6, RAPTOR excels across multiple metrics. For ROUGE-L, it surpasses BM25 and DPR by 7.3 and 2.7 points, respectively. In other metrics like BLEU-1, BLEU-4, and METEOR, RAPTOR outperforms BM25 and DPR by margins ranging from 1.7 to 5.8 and 0.7 to 2.1 points, respectively.

---

[2]For the DPR experiments in Tables 1 and 2, we used the `dpr-multiset-base` model as opposed to `dpr-single-nq-base` which was used in rest of the experiments done earlier. This decision was based on the performance observed in Karpukhin et al. (2020), where `dpr-multiset-base` showed superior results.

Table 2: **QuALITY and QASPER Performance With + Without RAPTOR:** Performance comparison across the QuALITY and QASPER datasets of various retrieval methods (SBERT, BM25, DPR) with and without RAPTOR. UnifiedQA-3B is used as the language model. RAPTOR outperforms baselines of each respective retrieval method for both datasets.

| Model | Accuracy (QuALITY) | Answer F1 (QASPER) |
|---|---|---|
| **SBERT with RAPTOR** | **56.6%** | **36.70%** |
| SBERT without RAPTOR | 54.9% | 36.23% |
| **BM25 with RAPTOR** | **52.1%** | **27.00%** |
| BM25 without RAPTOR | 49.9% | 26.47% |
| **DPR with RAPTOR** | **54.7%** | **32.23%** |
| DPR without RAPTOR | 53.1% | 31.70% |

Table 3: Controlled comparison of F-1 scores on the QASPER dataset, using three different language models (GPT-3, GPT-4, UnifiedQA 3B) and various retrieval methods. The column "Title + Abstract" reflects performance when only the title and abstract of the papers are used for context. RAPTOR outperforms the established baselines BM25 and DPR across all tested language models. Specifically, RAPTOR's F-1 scores are at least 1.8% points higher than DPR and at least 5.3% points higher than BM25.

| Retriever | GPT-3 F-1 Match | GPT-4 F-1 Match | UnifiedQA F-1 Match |
|---|---|---|---|
| Title + Abstract | 25.2 | 22.2 | 17.5 |
| BM25 | 46.6 | 50.2 | 26.4 |
| DPR | 51.3 | 53.0 | 32.1 |
| **RAPTOR** | **53.1** | **55.7** | **36.6** |

**Comparison to State-of-the-art Systems** Building upon our controlled comparisons, we examine RAPTOR's performance relative to other state-of-the-art models. As shown in Table 5, RAPTOR with GPT-4 sets a new benchmark on QASPER, with a 55.7% F-1 score, surpassing the CoLT5 XL's score of 53.9%.

In the QuALITY dataset, as shown in Table 7, RAPTOR paired with GPT-4 sets a new state-of-the-art with an accuracy of 82.6%, surpassing the previous best result of 62.3%. In particular, it outperforms CoLISA by 21.5% on QuALITY-HARD, which represents questions that humans took unusually long to correctly answer, requiring rereading parts of the text, difficult reasoning, or both.

For the NarrativeQA dataset, as represented in Table 6, RAPTOR paired with UnifiedQA sets

Table 4: Comparison of accuracies on the QuALITY dev dataset for two different language models (GPT-3, UnifiedQA 3B) using various retrieval methods. RAPTOR outperforms the baselines of BM25 and DPR by at least 2.0% in accuracy.

| Model | GPT-3 Acc. | UnifiedQA Acc. |
|---|---|---|
| BM25 | 57.3 | 49.9 |
| DPR | 60.4 | 53.9 |
| **RAPTOR** | **62.4** | **56.6** |

Table 5: Results on F-1 Match scores of various models on the QASPER dataset.

| Model | F-1 Match |
|---|---|
| LongT5 XL (Guo et al., 2022) | 53.1 |
| CoLT5 XL (Ainslie et al., 2023) | 53.9 |
| **RAPTOR + GPT-4** | **55.7** |

a new state-of-the-art METEOR score. When compared to the recursively summarizing model by Wu et al. (2021), which also employs UnifiedQA, RAPTOR outperforms it on all metrics. While Wu et al. (2021) rely solely on the summary in the top root node of the tree structure, RAPTOR benefits from its intermediate layers and clustering approaches, which allows it to capture a range of information, from general themes to specific details, contributing to its overall strong performance.

## 4.1 CONTRIBUTION OF THE TREE STRUCTURE

We examine the contribution of each layer of nodes to RAPTOR's retrieval capabilities. We hypothesized that upper nodes play a crucial role in handling thematic or multi-hop queries requiring a broader understanding of the text.

Table 6: Performance comparison on the NarrativeQA dataset across multiple models, focusing on four metrics: ROUGE-L, BLEU-1, BLEU-4, and METEOR. RAPTOR, when paired with UnifiedQA 3B, not only surpasses retrieval methods like BM25 and DPR but also sets a new state-of-the-art in the METEOR metric.

| Model | ROUGE-L | BLEU-1 | BLEU-4 | METEOR |
|---|---|---|---|---|
| BiDAF (Kočiský et al., 2018) | 6.2 | 5.7 | 0.3 | 3.7 |
| BM25 + BERT (Mou et al., 2020) | 15.5 | 14.5 | 1.4 | 5.0 |
| Recursively Summarizing Books (Wu et al., 2021) | 21.6 | 22.3 | 4.2 | 10.6 |
| Retriever + Reader (Izacard & Grave, 2022) | **32.0** | **35.3** | **7.5** | 11.1 |
| **RAPTOR + UnifiedQA** | 30.8 | 23.5 | 6.4 | **19.1** |

Table 7: Accuracies of the QuALITY dataset on both the overall test set and the more challenging hard subset. GPT-4 with RAPTOR sets a new state-of-the-art.

| Model | Accuracy | |
|---|---|---|
| | Test Set | Hard Subset |
| Longformer-base (Beltagy et al., 2020) | 39.5 | 35.3 |
| DPR and DeBERTaV3-large (Pang et al., 2022) | 55.4 | 46.1 |
| CoLISA (DeBERTaV3-large) (Dong et al., 2023a) | 62.3 | 54.7 |
| **RAPTOR + GPT-4** | **82.6** | **76.2** |

Table 8: Performance of RAPTOR when querying different tree layers for Story 1 from the QuALITY dataset. Columns represent different starting points (highest layer) and rows represent different numbers of layers queried.

| Layers Queried / Start Layer | Layer 0 (Leaf Nodes) | Layer 1 | Layer 2 |
|---|---|---|---|
| 1 layer | 57.9 | 57.8 | 57.9 |
| 2 layers | - | 52.6 | 63.15 |
| 3 layers | - | - | **73.68** |

We validated this hypothesis both quantitatively and qualitatively. We present qualitative analysis in appendix G. To quantitatively understand the contribution of the upper-level nodes, we used stories from the QuALITY dataset. The RAPTOR tree is built for each of these stories, as described in Section 3. However, during retrieval, we limit the search to different subsets of layers. For example, we exclusively retrieve from the leaf nodes and each upper layer, as well as from different contiguous subsets of the layers. We show findings specific to one story in Table 8, revealing that a full-tree search, utilizing all layers, outperformed retrieval strategies that focused only on specific layers.

These findings highlight the importance of the full tree structure in RAPTOR. By providing both the original text and higher-level summaries for retrieval, RAPTOR can effectively handle a wider range of questions, from higher-order thematic queries to detail-oriented questions. Detailed results for additional stories and an ablation study on layer contributions can be found in Appendix I.

## 5 CONCLUSION

In this paper, we have presented RAPTOR, a novel tree-based retrieval system that augments the parametric knowledge of large language models with contextual information at various levels of abstraction. By employing recursive clustering and summarization techniques, RAPTOR creates a hierarchical tree structure that is capable of synthesizing information across various sections of the retrieval corpora. During the query phase, RAPTOR leverages this tree structure for more effective retrieval. Our controlled experiments demonstrated that RAPTOR not only outperforms traditional retrieval methods but also sets new performance benchmarks on several question-answering tasks.

## 6 REPRODUCIBILITY STATEMENT

**Language Models for QA and Summarization** Four language models are used in our RAPTOR experiments: GPT-3 and GPT-4 for QA tasks, and GPT-3.5-turbo for summarization. The gpt-3, gpt-4, and gpt-3.5-turbo models can be accessed via API calls (OpenAI API). UnifiedQA, which is used for QA tasks, is publicly available at Hugging Face.

**Evaluation Datasets** The three evaluation datasets used in our experiments—QuALITY, QASPER, and NarrativeQA—are all publicly accessible. These datasets ensure that the retrieval and QA tests conducted in this study can be replicated.

**Source Code** We have released the code of RAPTOR at https://github.com/parthsarthi03/raptor.

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

# A   SCALABILITY AND COMPUTATIONAL EFFICIENCY OF THE TREE-BUILDING PROCESS

To assess the computational efficiency and cost-effectiveness of RAPTOR's tree-building process, we conducted experiments on a consumer-grade laptop, specifically an Apple M1 Mac with 16GB of RAM. These experiments aimed to demonstrate the scalability and feasibility of RAPTOR on typical hardware. We varied the context length from 12,500 to 78,000 tokens and measured both the token expenditure and the time required to complete the tree-building process, from initial splitting and embedding to the construction of the final root node.

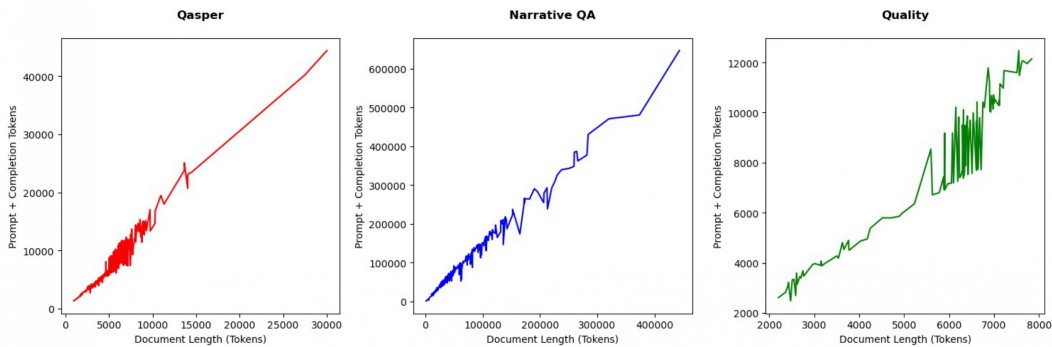

Figure 5: Token cost as a function of document length for QASPER, NarrativeQA, and QuALITY. RAPTOR tree construction costs scale linearly with document length for each of the datasets.

**Token Expenditure**   We empirically investigated the relationship between the initial document length and the total number of tokens expended during the tree-building process, which includes both the prompt and completion tokens. The document lengths varied significantly across the three

datasets examined: QuALITY, QASPER, and NarrativeQA. Figure 5 illustrates a clear linear correlation between the initial document length and the total token expenditure, emphasizing that RAPTOR maintains a linear token scaling regardless of document complexity or length.

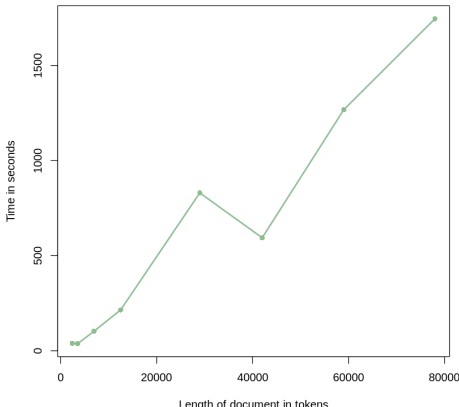

Figure 6: Build time as a function of document length for documents of up to 80,000 tokens. RAPTOR tree construction time scales linearly with document length for each of the datasets.

**Build Time**    We also empirically observed a consistent linear trend between the document length and the build time, as shown in Figure 6. This suggests that RAPTOR scales linearly in terms of time, making it a viable solution for efficiently processing large corpora of varying lengths.

**Conclusion**    Overall, our empirical results indicate that RAPTOR scales both in terms of tokens expended and build time. Even as the complexity and volume of the input text grow, the cost of constructing the tree scales predictably and linearly. This demonstrates that RAPTOR is computationally efficient and well-suited for processing large and diverse corpora.

## B    ABLATION STUDY ON CLUSTERING MECHANISM IN RAPTOR

To assess the effectiveness of the clustering mechanism in our RAPTOR approach, we conducted an ablation study on the QuALITY dataset. This study compares RAPTOR's performance with a balanced tree-style encoding and summarization of contiguous chunks, in contrast to our standard clustering method.

### B.1    METHODOLOGY

Both configurations in this ablation study utilized SBERT embeddings and UnifiedQA to maintain consistency in retrieval. For RAPTOR, we employed our typical clustering and summarization process. In contrast, the alternative setup involved creating a balanced tree by recursively encoding and summarizing contiguous text chunks. We determined the window size for this setup based on the average cluster size observed in RAPTOR, which is approximately 6.7 nodes. Hence, we chose a window size of 7 nodes. The collapsed tree approach was applied for retrieval in both models.

### B.2    RESULTS & DISCUSSION

The results of the ablation study are presented in table 9. The results from this ablation study clearly indicate an improvement in accuracy when employing RAPTOR's clustering mechanism over the recency-based tree approach. This finding substantiates our hypothesis that the clustering strategy in RAPTOR is more effective in capturing homogeneous content for summarization, thereby enhancing the overall retrieval performance.

Table 9: Ablation study results comparing RAPTOR with a recency-based tree approach

| Configuration | Accuracy |
|---|---|
| **RAPTOR + SBERT embeddings + UnifiedQA** | **56.6%** |
| Recency-based tree + SBERT embeddings + UnifiedQA | 55.8% |

## C  DATASET STATISTICS AND COMPRESSION RATIOS

The average ratio of the summary length to the sum of child node lengths across all datasets is 0.28, indicating a 72% compression rate. On average, the summary length is 131 tokens, and the average child node length is 86 tokens. Below are the detailed statistics for all three datasets:

Table 10: Statistics of Average Summary Length and Child Node Length Across Datasets

| Dataset | Avg. Summary Length (tokens) | Avg. Child Node Text Length (tokens) | Avg. # of Child Nodes Per Parent | Avg. Compression Ratio (%) |
|---|---|---|---|---|
| All Datasets | 131 | 85.6 | 6.7 | .28 |
| QuALITY | 124.4 | 87.9 | 5.7 | .28 |
| NarrativeQA | 129.7 | 85.5 | 6.8 | .27 |
| QASPER | 145.9 | 86.2 | 5.7 | .35 |

## D  SUMMARIZATION PROMPT

Table 11 shows the prompt used for summarization.

Table 11: Prompt for Summarization

| Role | Content |
|---|---|
| system | You are a Summarizing Text Portal |
| user | Write a summary of the following, including as many key details as possible: {context}: |

## E  HALLUCINATION ANALYSIS

To assess the quality and accuracy of the summarizations within our RAPTOR model, we conducted an analysis focusing on hallucinations in the generated summaries. The summaries were generated by `gpt-3.5-turbo` and subsequently annotated to quantify the rates of hallucinations, to examine whether such inaccuracies propagate to parent nodes, and to evaluate their impact on question-answering (QA) tasks.

### E.1  METHODOLOGY

We randomly sampled 150 nodes across 40 stories and evaluated them for hallucinations. This sampling strategy provides a broad view of the model's performance across different contexts. Each node was annotated by hand, and determined if it contained a hallucination.

### E.2  FINDINGS

Out of the 150 nodes sampled, 4% (6 nodes) contained some form of hallucination. Most commonly, these hallucinations originated from the model adding minor information possibly from its training data that was not present in the text being summarized, or from incorrectly extrapolating some information when creating the summary.

**Example:**

*Text of the child nodes:*

> "And you will come with me to my people? We may live here among them, and you will be a great warrior–oh, when Jor dies you may even be chief, for there is none so mighty as my warrior..."But your father will not permit it–Jor, my father, High Chief of the Galus, will not permit it, for like me you are cos-ata-lo. Oh, Co-Tan, if we but could!... Bradley noticed that she spoke in English–broken English like Co-Tan's but equally appealing.

*Summary found in the parent of that node:*

> The protagonist, Bradley, is being asked by Co-Tan to stay with her people and become a great warrior, but he refuses and must return to his own country. Tom Billings of Santa Monica arrives and tells them he came to search for a man named Bowen J. Tyler, Jr. Ajor, Co-Tan's sister, is excited about the possibility of going to Tom's country to see strange and wonderful things...

The hallucination here is that the summary states that Jr. Ajor and Co-Tan are sisters, but does not explicitly mention or imply this.

Upon reviewing all parent nodes, we found that hallucinations did not propagate to higher layers. Generally, the hallucinations were minor and did not alter the thematic interpretation of the text.

### E.3 IMPACT ON QA TASKS

In our findings, hallucinations had no discernible impact on the performance of QA tasks. This suggests that hallucination is not a major concerns for the summarization component in our RAPTOR architecture.

## F PSEUDOCODE FOR RETRIEVAL METHODS

---
**Algorithm 1** Tree Traversal Algorithm
---

**function** TRAVERSETREE(tree, query, $k$)
    $S_{current} \leftarrow$ tree.layer[0]
    **for** layer in range(tree.num_layers) **do**
        $top_k \leftarrow []$
        **for** node in $S_{current}$ **do**
            $score \leftarrow$ dot_product(query, node)
            top_k.append((node, score))
        **end for**
        $S_{layer} \leftarrow$ sorted(top_k)[:k].nodes
        $S_{current} \leftarrow S_{layer}$
    **end for**
    **return** $S_0 \cup S_1 \cup S_2 \cup \ldots \cup S_k$
**end function**

---

## G QUALITATIVE ANALYSIS

To qualitatively examine RAPTOR's retrieval process, we test it on thematic, multi-hop questions about a 1500-word version of the fairytale Cinderella. We compare the context retrieved by RAPTOR with the context retrieved by Dense Passage Retrieval (DPR). Figure 4 in the main paper details the retrieval process within RAPTOR's tree structure for two questions. The nodes that RAPTOR selects for each question are highlighted, while the leaf nodes that DPR selects for the same question are indicated with arrows. This comparison illustrates the advantage of RAPTOR's tree structure. RAPTOR selects nodes from different layers depending on the level of granularity required by the

---

**Algorithm 2** Collapsed Tree Algorithm

---

**function** COLLAPSEDTREE(tree, query, $k$, max_tokens)
    tree ← flatten(tree)                                  ▷ Flatten tree into 1D
    top_nodes ← []
    **for** node in tree **do**
        top_nodes.append((node, dot_product(query, node)))
    **end for**
    top_nodes ← sorted(top_nodes)
    result ← []
    total_tokens ← 0
    **for** node in top_nodes **do**
        **if** total_tokens + node.token_size < max_tokens **then**
            result.append(node)
        **end if**
        total_tokens ← total_tokens + node.token_size
    **end for**
    **return** result
**end function**

---

| Question: What is the central theme of the story? | |
|---|---|
| **RAPTOR** | Fairy Godmother helps Cinderella attend a ball by transforming her rags... Cinderella impresses the Prince at the ball... she loses track of time and has to run home alone in the darkness. The Prince is unable to find Cinderella and goes in search of her ... She forgave her sisters, and treated them always very kindly, and the Prince had great cause to be glad that he had found the glass slipper. |
| **DPR** | Two mice were turned into footmen; four grasshoppers into white horses. Next, the Fairy touched Cinderella's rags, and they became rich satin robes, trimmed with point lace. ... Alas! the clock had struck Eleven. She found no coach, only a pumpkin, and the rat and mice ran quickly away when they saw her; while all her fine dress turned to rags, and she had to run home alone... They told her a beautiful Princess had been at the ball, with whom the Prince was delighted. They did not know it was Cinderella. |
| Question: How does Cinderella find a happy ending? | |
| **RAPTOR** | The story of Cinderella involves her godmother, a fairy, who transforms a pumpkin into a grand coach with her wand and allows Cinderella to attend the ball. However, Cinderella must return home before the clock strikes eleven or her dress will turn back into rags... Cinderella impresses the Prince at the ball but leaves before he can find out who she is... The Prince searched for the owner of a lost glass slipper and found it belonged to Cinderella. She forgave her sisters and the Prince was glad to have found her. |
| **DPR** | the clock had struck Eleven... The Prince was very much surprised when he missed Cinderella again, and leaving the ball, went in search of her... Fairy touched Cinderella's rags, and they became rich satin robes, trimmed with point lace... Her old shoes became a charming pair of glass slippers, which shone like diamonds. "Now go to the ball, my love," she said, "and enjoy yourself. But remember, you must leave the room before the clock strikes eleven. If you do not your dress will return to its original rags." |

Table 12: Relevant excerpts from text retrieved by RAPTOR and DPR for the questions on the fairytale Cinderella.

question at hand. Further, the information that would be retrieved by DPR is more often than not included in the context retrieved by RAPTOR, either directly as a leaf node or indirectly as part of a summary from a higher layer.

"The first question we examine is "How does Cinderella find a happy ending?", a multi-hop question best answered by synthesizing information from various text segments. To control for the language model's potential familiarity with the Cinderella story, we instructed it to rely solely on the retrieved information for its answers. Table 12 shows the text retrieved by both RAPTOR and DPR for this question. RAPTOR's context succinctly describes Cinderella's journey to happiness, while DPR's leaf nodes primarily focus on her initial transformation. The difference in retrieved information

significantly impacts downstream tasks. When GPT-4 is provided with RAPTOR's context, it generates a detailed answer: "Cinderella finds a happy ending when the Prince searches for the owner of the lost glass slipper and discovers it belongs to Cinderella. They eventually marry, transforming Cinderella's life for the better." In contrast, using DPR's context, GPT-4 states: "Based on the given context, it is not possible to determine how Cinderella finds a happy ending, as the text lacks information about the story's conclusion."

The second question we examine is "What is the central theme of the story?", a thematic question that requires holistic understanding of the entire text. The text retrieved by RAPTOR and DPR for this question is shown in Table 12. The text retrieved by RAPTOR contains short descriptions of all the major parts of the story, whereas the text retrieved by DPR contains detailed descriptions of a narrow subset of the story. Again, the difference in retrieval mechanisms affects the performance of GPT-4 when answering the question. Given DPR's context, it outputs "The central theme of the story is transformation and the power of inner beauty, as Cinderella, a kind and humble girl, is magically transformed into a beautiful princess, capturing the attention and admiration of the Prince and others at the ball." This answer only takes into account the first portion of the story, up until Cinderella first meets the prince. In contrast, given RAPTOR's context, GPT-4 outputs "The central theme of the story is transformation and overcoming adversity, as Cinderella, with the help of her Fairy Godmother, transforms from a mistreated and downtrodden girl into a beautiful and confident young woman who ultimately finds happiness and love with the Prince." This is a more complete answer, demonstrating a comprehensive understanding of the story.

This qualitative analysis indicates that RAPTOR outperforms prior retrieval mechanisms because the information that it retrieves is more relevant and exhaustive, allowing for better performance on downstream tasks.

We also created a 2600-word story along with questions about its narrative and theme. An excerpt from the story is present below and the full PDF of this story is linked here. For questions like "What is the central theme of the story?", an upper-level node is retrieved which includes the sentence: "This story is about the power of human connection... inspiring and uplifting each other as they pursued their passions." This summary, not explicitly present in the original text, almost directly answers the question.

**Excerpt from "The Eager Writer":**

> "Ethan's passion for writing had always been a part of him. As a child, he would often scribble stories and poems in his notebook, and as he grew older, his love for writing only intensified. His evenings were often spent in the dim light of his room, typing away at his laptop. He had recently taken a job as a content writer for an online marketing firm to pay the bills, but his heart still longed for the world of storytelling. However, like many aspiring writers, he struggled to find a foothold in the industry. He took a job as a content writer for an online marketing firm, but it was growing increasingly evident to him that this was not the path he wanted to pursue. It was during this time that he stumbled upon the Pathways app. The app offered a platform for people in similar professions to connect and share knowledge, and he saw it as an opportunity to finally connect with others who shared his passion for writing. Ethan saw an opportunity to meet others who shared his passion and could offer guidance and mentorship. He quickly signed up and was surprised by the number of writers he found on the platform, from well establish professionals to beginners just starting out in the business."

## H NARRATIVEQA EVALUATION SCRIPT

We made several modifications to AllenNLP's evaluation script[3] to better fit our evaluation needs:

- **Added Smoothing:** Smoothing was incorporated to handle cases where BLEU score is zero, due to no n-gram matches occurring in the reference text. A BLEU score of zero skews the results, leading to an overly harsh evaluation for rare or novel phrases. By adding

---

[3]docs.allennlp.org/models/main/models/rc/tools/narrativeqa/

a smoothing function, we prevent the BLEU scores from dropping to zero, providing a more fair evaluation.

- **Modified BLEU-4 Weighting:** The original script applied a weight of 1 to the highest order n-gram (4-gram) and 0 to the rest in its BLEU-4 calculation (i.e., weights=(0, 0, 0, 1)). This approach may overly focus on 4-gram matches while neglecting lower-order matches. To provide a more balanced evaluation, we evenly distributed the weight across all n-gram levels, changing the weights for the BLEU-4 calculation to (0.25, 0.25, 0.25, 0.25).

- **Tokenization before Mapping in METEOR Calculation:** The original script utilized a simple split and map method for METEOR calculation. We fixed this by first tokenizing the text and then mapping the tokens. This amendment improves the accuracy of the METEOR calculation by taking into account the correct linguistic boundaries of words.

## I  ANALYSIS OF DIFFERENT LAYERS ON RAPTOR'S PERFORMANCE

### I.1  HOW DO DIFFERENT LAYERS IMPACT PERFORMANCE ?

In this section, we present a detailed breakdown of RAPTOR's retrieval performance when querying different layers of the hierarchical tree structure for various stories. These tables validate the utility of RAPTOR's multi-layered structure for diverse query requirements.

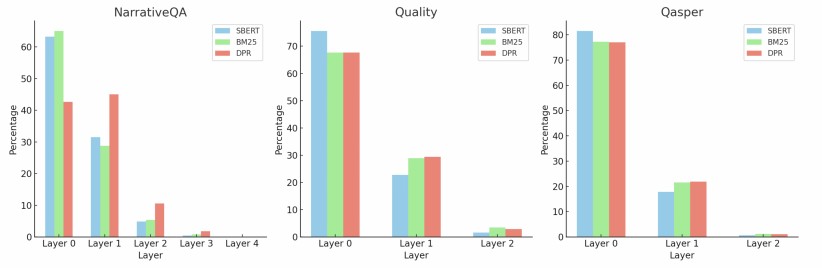

Figure 7: Histogram showing the percentage of nodes retrieved from different layers of the RAPTOR tree across three datasets (NarrativeQA, Quality, and Qasper) using three retrievers (SBERT, BM25, and DPR). The data indicate that a substantial portion of the nodes contributing to the final retrieval comes from non-leaf layers, with a notable percentage from the first and second layers, highlighting the importance of RAPTOR's hierarchical summarization in the retrieval process.

Table 13: Performance of RAPTOR when querying different layers of the tree for Story 2.

| Layers Queried / Start Layer | Layer 0 (Leaf Nodes) | Layer 1 | Layer 2 |
|---|---|---|---|
| 1 layer | 58.8 | 47.1 | 41.1 |
| 2 layers | - | **64.7** | 52.9 |
| 3 layers | - | - | 47.1 |

Table 14: Performance of RAPTOR when querying different layers of the tree for Story 3.

| Layers Queried / Start Layer | Layer 0 (Leaf Nodes) | Layer 1 | Layer 2 |
|---|---|---|---|
| 1 layer | 66.6 | 61.1 | 61.1 |
| 2 layers | - | 66.6 | 66.6 |
| 3 layers | - | - | **83.3** |

### I.2  WHICH LAYERS DO RETRIEVED NODES COME FROM ?

We further conduct an ablation study across all three datasets and across three different retrievers with RAPTOR with the collapsed tree retrieval to examine the layers from which the retrieved nodes

Table 15: Performance of RAPTOR when querying different layers of the tree for Story 4.

| Layers Queried / Start Layer | Layer 0 (Leaf Nodes) | Layer 1 |
|---|---|---|
| 1 layer | **94.7** | 84.2 |
| 2 layers | - | 89.4 |

Table 16: Performance of RAPTOR when querying different layers of the tree for Story 5.

| Layers Queried / Start Layer | Layer 0 (Leaf Nodes) | Layer 1 |
|---|---|---|
| 1 layer | 57.9 | 47.3 |
| 2 layers | - | **68.4** |

originate. We observe that between 18.5% to 57% of the retrieved nodes come from non-leaf nodes. As illustrated in Figure 7, the retrieval pattern across layers reveals the importance of RAPTOR's multi-layered tree structure. Notably, a significant percentage of the nodes retrieved by RAPTOR using the DPR retriever for the NarrativeQA dataset come from the first and second layers of the tree, as opposed to the leaf nodes. This pattern is consistent across the other datasets and retrievers, albeit with varying percentages.

Table 17: Percentage of nodes from non-leaf nodes across different datasets and retrievers

| Dataset | DPR | SBERT | BM25 |
|---|---|---|---|
| NarrativeQA | 57.36% | 36.78% | 34.96% |
| Quality | 32.28% | 24.41% | 32.36% |
| Qasper | 22.93% | 18.49% | 22.76% |

Table 18: Percentage of nodes from different layers with DPR as the retriever

| Layer | NarrativeQA | Quality | Qasper |
|---|---|---|---|
| 0 | 42.64% | 67.71% | 77.07% |
| 1 | 45.00% | 29.43% | 21.88% |
| 2 | 10.57% | 2.85% | 1.05% |
| 3 | 1.78% | - | - |
| 4 | 0.003% | - | - |

Table 19: Percentage of nodes from different layers with SBERT as the retriever

| Layer | NarrativeQA | Quality | Qasper |
|---|---|---|---|
| 0 | 63.22% | 75.59% | 81.51% |
| 1 | 31.51% | 22.78% | 17.84% |
| 2 | 4.85% | 1.63% | 0.65% |
| 3 | 0.42% | - | - |

Table 20: Percentage of nodes from different layers with BM25 as the retriever

| Layer | NarrativeQA | Quality | Qasper |
|---|---|---|---|
| 0 | 65.04% | 67.64% | 77.24% |
| 1 | 28.79% | 28.85% | 21.57% |
| 2 | 5.36% | 3.51% | 1.19% |
| 3 | 0.81% | - | - |

