# OpenReview forum: "RAPTOR: Recursive Abstractive Processing for Tree-Organized Retrieval"
_ICLR.cc/2024/Conference — ICLR 2024 poster_

### Official Review · Reviewer_DRXQ · 2023-10-30

**Soundness:** 3 good
**Presentation:** 3 good
**Contribution:** 3 good
**Rating:** 8
**Confidence:** 3

**Summary:**

In the context of reML, this paper proposes to extend the support documents by generating summaries at different level of granularity (full document to the original text). Summaries are based on a clustered set of chunks of text within a UMAP representation space (which is the novelty in this paper), and the process is recursive. This new set of passages is used as a new dataset, with two strategies for searching: either considering each passage independently, or using a hierarchy to bias the search. The latter strategy is working much worse, and hence the authors focus on the first. The model perform experiments with this strategy, using GPT-3.5/4 to generate the final answer, and comparing to other summarization-based approaches on NarrativeQA, Qasper and QuALITY - and improve the results on the three datasets (especially QuALITY).

**Strengths:**

While the model novelty is low -- instead of using contiguous segments for summarization, the authors propose to segments cluster by similarity -- this simple idea improves results on three different datasets. The different experiments conducted show that the hierarchy is only useful because it produces summaries at different granularity levels (but it does not help in searching for the right passages).

**Weaknesses:**

It would be good to see the performance using open-domain pre-trained models (e.g. LLAMA or other ones), to ensure a more reproducible research.

There is no experiment on the impact of the clustering algorithm (what is the difference when not using UMAP? when using more or less neighbors?).

**Questions:**

- Please give examples of generated summaries.

- It would be great to have some insight on the impact of the clustering algorithm, and how the performance degrades with summary quality

---

> ### Author Response · Authors · 2023-11-22
> **Response to Reviewer DRXQ**
>
> Thank you for your insightful review and for the new experiments you suggested! Please see our detailed response below.
>
>
> > **It would be good to see the performance using open-domain pre-trained models (e.g. LLAMA or other ones), to ensure a more reproducible research.**
>
> We appreciate the reviewer's suggestion to use open-domain pre-trained models to enhance the reproducibility of our research. Note that our existing results use the UnifiedQA model, which is an open-source model. Furthermore, as per the reviewer’s suggestion, we conducted additional experiments using the instruction-tuned Llama2-7B-chat model. We include the results below:
>
> | Model                | Answer F1  |
> |----------------------|------------|
> | **SBERT with RAPTOR**| **31.0%**  |
> | SBERT without RAPTOR | 25.5%      |
>
> These results show that RAPTOR significantly improves performance when using open-domain pre-trained models. To further enable reproducibility, we also release all of our code as supplementary materials.
>
> > **There is no experiment on the impact of the clustering algorithm (what is the difference when not using UMAP? when using more or less neighbors?).**
>
> We thank the reviewer for their insightful comments regarding the need for an in-depth analysis of the clustering algorithm used in our RAPTOR model. In our current manuscript, we have focused on demonstrating the overall efficacy of RAPTOR and its performance relative to existing retrieval methods. We agree that a detailed examination of the clustering algorithm, including the use of UMAP and the choice of the number of neighbors, will lead to a deeper understanding of RAPTOR's performance. We also believe that better and more optimized clustering methods could further improve the performance of RAPTOR, and would welcome future work in this area.
>
> To examine the impact of the clustering algorithm, we conducted a new ablation study comparing the clustering mechanism in RAPTOR against a baseline in which we construct the RAPTOR tree using contiguous chunks of text. We evaluate the QuALITY dataset, and the results are as follows:
>
> | Configuration                        | Accuracy   |
> |--------------------------------------|------------|
> | **RAPTOR + SBERT embeddings + UnifiedQA** | **56.6%**      |
> | Recency-based tree + SBERT embeddings + UnifiedQA | 55.8% |
>
> As shown above, the clustering mechanism in RAPTOR provides an improvement in accuracy over the recency-based tree approach. This supports the hypothesis that clustering enables more effective capture of homogeneous content for summarization and enhances performance.
>
> We have updated the paper with the results of this new experiment. Thank you for this suggestion, which we believe strengthens the work.
>
> ### Questions
>
> > **Please give examples of generated summaries.**
>
> To address the reviewer’s request and provide greater transparency, we will release all RAPTOR tree structures (which contain all summaries) as binaries. We hope that this also enables further analysis by external researchers.
>
> > **It would be great to have some insight on the impact of the clustering algorithm, and how the performance degrades with summary quality.**
>
> We thank the reviewer for their valuable suggestion to explore the impact of the clustering algorithm (and we have described new experiments above), as well as the effect of summary quality on performance. Note that, in our initial setup, the tree summaries were crafted using the GPT-3.5 Turbo model.
>
> In response to the reviewer’s suggestion for further insights, we have conducted a new experiment to assess how performance varies as a function of summary quality. For this purpose, we produced tree summaries using the Llama2-7B-chat model for a subset of 50 passages from the QASPER dataset. The results of this experiment (shown below) are somewhat surprising: while we anticipated a potential performance degradation when shifting from GPT-3.5 to Llama2-7B-chat, the performance impact was in fact negligible. This suggests that even with summaries of lower quality, RAPTOR remains largely unaffected. Here are the results:
>
> Both experiments used the Llama2-7B-chat model for question-answering
> | Configuration                        | Answer F1   |
> |--------------------------------------|------------|
> | RAPTOR + GPT-3.5 Turbo Tree Summaries | 31.0  |
> | RAPTOR + Llama2-7B-chat Tree Summaries| 28.8 |
>
> The result above suggests that RAPTOR can potentially be applied using far fewer computational resources without compromising performance, and we have added these results to the revised paper. Thank you for suggesting that we explore this aspect of our method!

---

> > ### Author Response · Authors · 2023-11-23
> >
> > Thank you once again for your valuable feedback! As we approach the end of our discussion period, we wanted to check in to see if our previous response has addressed your concerns and answered your questions. If you have any additional questions or if there are any concerns that we haven't yet addressed, please let us know and we would be happy to answer them.

---

### Official Review · Reviewer_PvjZ · 2023-11-02

**Soundness:** 3 good
**Presentation:** 3 good
**Contribution:** 4 excellent
**Rating:** 8
**Confidence:** 3

**Summary:**

The paper introduces a new information retrieval mechanism for retrieval augmented generation (RAG). Traditionally, retrieval in RAG is done rather flatly by indexing text-chunks and then retrieving top-k. In the new approach, the authors create a hierarchical encoding. They do this by:

1. Encoding original text-chunks.
2. Clustering the (encoding,text-chunk) pairs.
3. Summarizing the clustered text-chunks into a single text-chunk.
4. Encoding the summary text-chunks to get new (encoding,text-chunk) pairs .
5. Going to step 2 and repeating the process until the root encoding is created.

They also propose two ways to query/retrieve from the hierarchical encoding - one method involving plain top-k retrieval from flattened tree, and another method starts from the root encoding and explores the tree of encodings via beam search. In both cases, information at various levels of granularity can be retrieved.

**Strengths:**

1. Overall an elegant, well-motivated, and seemingly novel (to my knowledge) approach. It can set up a new paradigm baseline for RAG that can incite future research on refining the general idea.

2. Decent performance compared to other paradigmatic approaches for retrieval such as DPR.

**Weaknesses:**

1. One limitation seems to be that the approach requires recursive summarization with an LLM which can add to computational expense (could be also good to share the trade-off).


2. While I get the theoretical intuition of clustering (to prevent information loss, by clustering more homogenous content for summarization), it would have been nice to have an empirical demonstration of the effectiveness of clustering. A possible ablation could be: what if we took a balanced tree-style encoding/summarization and simply recursively encoded contiguous chunks (instead of clustering)?


3. There is some precedence for using hierarchical retrieval (even though there are crucial differences) [1,2] that could be cited and discussed for better contextualization in the literature.


[1] Dense Hierarchical Retrieval for Open-domain Question Answering - Liu et al. EMNLP Findings 2021

[2] Hybrid Hierarchical Retrieval for Open-Domain Question Answering - Arivazhagan et al. ACL Findings 2023

**Questions:**

1. It could be good to add a pseudo-code for beam tree-based retrieval and collapsed-tree-based retrieval.
2. The 20% accuracy boost claim (while technically correct) may be a bit misleading given prior works do not use GPT4 (if I understand correctly). So most of the boost most likely comes from GPT4 rather than RAPTOR.

---

> ### Author Response · Authors · 2023-11-22
> **Response to Reviewer PvjZ [1/2]**
>
> Thank you for your thoughtful review and helpful suggestions! Please see our response below:
>
>
> > **One limitation seems to be that the approach requires recursive summarization with an LLM which can add to computational expense (could be also good to share the trade-off).**
>
> We acknowledge the reviewer's point regarding the computational expense incurred by the recursive summarization process in our RAPTOR method. In response to this concern, we conducted an ablation study to understand the contribution of each layer in RAPTOR's tree structure to the overall performance. This study is aimed at identifying potential optimizations to mitigate computational costs.
>
> Our analysis, presented in the tables below, reveals that the majority of summary nodes contributing to the answers are from the first (non-leaf) layer of the RAPTOR tree. This finding suggests that, in scenarios where computational resources are constrained, it might be practical to limit the recursive clustering process to a single layer. By doing so, we can significantly reduce computational expenses while still capturing a substantial portion of the relevant information for effective retrieval.
>
> We have incorporated this analysis and recommendation into our paper, providing readers with insights into how they can adapt RAPTOR to their computational constraints.
>
> ####  % of nodes from different layer with DPR as the retriever
> | Layer | NarrativeQA | Quality | Qasper |
> |-------|-------------|---------|--------|
> | 0     | 42.64       | 67.71   | 77.07  |
> | 1     | 45.00       | 29.43   | 21.88  |
> | 2     | 10.57       | 2.85    | 1.05   |
> | 3     | 1.78        | -       | -      |
> | 4     | 0.003       | -       | -      |
>
> #### % of nodes from different layer with SBERT as the retriever
> | Layer | NarrativeQA | Quality | Qasper |
> |-------|-------------|---------|--------|
> | 0     | 63.22       | 75.59   | 81.51  |
> | 1     | 31.51       | 22.78   | 17.84  |
> | 2     | 4.85        | 1.63    | 0.65   |
> | 3     | 0.42        | -       | -      |
>
> #### % of nodes from different layer with BM25 as the retriever
> | Layer | NarrativeQA | Quality | Qasper |
> |-------|-------------|---------|--------|
> | 0     | 65.04       | 67.64   | 77.24  |
> | 1     | 28.79       | 28.85   | 21.57  |
> | 2     | 5.36        | 3.51    | 1.19   |
> | 3     | 0.81        | -       | -      |

---

> ### Author Response · Authors · 2023-11-22
> **Response to Reviewer PvjZ [2/2]**
>
> > **While I get the theoretical intuition of clustering (to prevent information loss, by clustering more homogenous content for summarization), it would have been nice to have an empirical demonstration of the effectiveness of clustering. A possible ablation could be: what if we took a balanced tree-style encoding/summarization and simply recursively encoded contiguous chunks (instead of clustering)?**
>
> We appreciate the reviewer's suggestion for an empirical demonstration of the effectiveness of clustering in our RAPTOR model. We conducted the proposed ablation study on the QuALITY dataset, comparing the performance of RAPTOR with a balanced tree-style encoding and summarization of contiguous chunks, rather than clustering.
>
> In this ablation, we maintained a consistent retrieval setup using SBERT embeddings and UnifiedQA across both configurations. For RAPTOR, we followed our standard methodology of clustering and summarization. In the alternative setup, we created a balanced tree structure by recursively encoding and summarizing contiguous chunks of text, setting the window size to the average cluster size observed in RAPTOR (approximately 6.7 nodes, thus a window size of 7 nodes). For both models, we employed the collapsed tree approach for retrieval. The results of this ablation are as follows:
>
> | Configuration                        | Accuracy   |
> |--------------------------------------|------------|
> | **RAPTOR + SBERT embeddings + UnifiedQA** | **56.6%**      |
> | Recency-based tree + SBERT embeddings + UnifiedQA | 55.8% |
>
> The findings from this study demonstrate that the clustering mechanism in RAPTOR provides an improvement in accuracy over the recency-based tree approach. This supports the hypothesis that clustering enables more effective capture of homogeneous content for summarization, enhancing retrieval performance. We have incorporated these results into the revised manuscript to further motivate our clustering approach.
>
> > **There is some precedence for using hierarchical retrieval (even though there are crucial differences) [1,2] that could be cited and discussed for better contextualization in the literature.**
>
> Thank you for pointing us to relevant literature on hierarchical retrieval. We have updated our paper to include these references!
>
> ### Questions
>
> > **It could be good to add a pseudo-code for beam tree-based retrieval and collapsed-tree-based retrieval.**
>
> We agree— we have added pseudo-code for both tree traversal methods to the appendix of our revised paper. For further transparency and to enable reproducibility, we have also provided all of our code as supplementary materials and intend to open-source it.
>
>
> > **The 20% accuracy boost claim (while technically correct) may be a bit misleading given prior works do not use GPT4 (if I understand correctly). So most of the boost most likely comes from GPT4 rather than RAPTOR.**
>
> We acknowledge the reviewer's concern regarding the 20% accuracy boost using RAPTOR in combination with GPT-4. Our findings indeed show that while GPT-4 contributes significantly to the overall performance, RAPTOR's unique retrieval approach provides an additional boost in accuracy. This is evident when we compare RAPTOR's performance with that of other retrieval methods paired with the GPT models. RAPTOR consistently outperforms these methods, indicating that its effectiveness is not solely due to the advanced capabilities of GPT models, but also due to the inherent advantages of the RAPTOR system.

---

> > ### Author Response · Authors · 2023-11-23
> >
> > Thank you once again for your valuable feedback! As we approach the end of the discussion period, we wanted to check in to see if our previous response has addressed your concerns and answered your questions. If you have any additional questions or if there are any concerns that we haven't yet addressed, please let us know and we would be happy to answer them.

---

> > > ### Comment · Reviewer_PvjZ · 2023-11-23
> > > **Thank you**
> > >
> > > Thank you for the feedback and further analyses. I do not have any deep concerns. For the computational trade-off, I was wondering more along the lines of the time for constructing the Tree nodes (say with SBERT + summarization) vs just flat encoding (with SBERT). But it is not too critical for my evaluation (given, I assume, that's a one-time expense unless the knowledge base is highly dynamic) and could be added to the final copy if possible. The analysis of the node usage percentage is also interesting and the suggestion for addressing the limitation makes sense.

---

> > > > ### Author Response · Authors · 2023-11-23
> > > >
> > > > The construction of the tree nodes is indeed a one-time computational expense. We are glad that the analysis of node usage percentage was insightful. As suggested, we will update the final copy of the paper to include a discussion on this computational aspect. Thank you once again for your valuable comments, which have helped us refine and strengthen our work.

---

### Official Review · Reviewer_YAgR · 2023-11-02

**Soundness:** 2 fair
**Presentation:** 2 fair
**Contribution:** 2 fair
**Rating:** 6
**Confidence:** 3

**Summary:**

In this paper, the authors propose a method called RAPTOR that retrieves chunks and summaries from tree-structured automatically created clusters based on the representation of SBERT. Since the clusters have a hierarchy, RAPTOR can consider high-level and low-level details of texts. To deal with the variable depth of the hierarchy, the cluster size is automatically decided by Gaussian Mixture with Bayesian Information Criterion (BIC) like x-means. The authors summarize each cluster by GPT-3.5-turbo from its elements, chunks, or summaries. When retrieving these chunks and summaries, the authors use two different methods. The first method, beam search, conducts a layer-wise traversal of the clusters beginning at the root nodes. The second method, collapsed tree, considers all nodes in the tree simultaneously. Experimental results show that RAPTOR improves the performance in various QA datasets.

**Strengths:**

- The proposed method RAPTOR can indirectly retrieve long texts by tracking tree-structured automatically created clusters.
- RAPTOR can decide the number of clusters automatically, and thus, it doesn't require manual tuning for creating the tree-structured clusters.
- RAPTOR can achieve better performances in QA compared with commonly used retrievers.

**Weaknesses:**

- Considering the information loss by summarization, the benefit of RAPTOR against enumerating possible concatenation of chunks is uncertain.
- Even if clustering is automatically done, how to segment texts into chunks is still left as a problem.
- When comparing model performances, the parameter sizes of models should be the same or similar. However, the paper compares models with totally different parameters. This is problematic from the viewpoint of fairness.
- How many retrieved instances are used for baseline models needs to be described in detail. Thus, judging the fairness of the comparison in the experiment is difficult.

**Questions:**

- How did you segment texts into chunks? If you use obvious segments in text like paragraphs, please state it clearly.
- How did you generate summaries by GPT-3.5-turbo? You should show the actually used prompt.
- What kind of embedding did you use for retrieving chunks and clusters? Did you use the centroids for each cluster or embedding of the summaries? Also, are these calculated by SBERT similar to the clustering step?
- This is related to the above weakness part. How many retrieved instances are used for baseline models?
- In the part, "Comparison to State-of-the-art Systems", you compared your RAPTOR with GPT-4 to LongT5 that has 3B of parameters and its variant, ColtT5. Considering that the detailed parameter size of GPT-4 is not released and the size of the baselines' parameters, this comparison is not fair. Did you adopt RAPTOR to LongT5 or ColtT5?
- Similar to the previous question, in Table 5, you should have adopted RAPTOR to the model with almost the same parameter size as that of baselines.

I can update my score based on your response.

---

> ### Author Response · Authors · 2023-11-22
> **Response to Reviewer YAgR [1/3]**
>
> We thank the reviewer for their insightful review! Please see our detailed responses below.
>
> > **Considering the information loss by summarization, the benefit of RAPTOR against enumerating possible concatenation of chunks is uncertain.**
>
> We appreciate the reviewer's comment on the potential information loss due to summarization in our RAPTOR model and have conducted additional experiments to directly address this concern. Our response is threefold:
>
> *Retention of Original Information in RAPTOR*: It is important to note that RAPTOR's leaf nodes retain the original, unsummarized text chunks. This design choice ensures that while higher-level nodes provide (potentially useful) summaries, the low-level details are preserved in the tree, mitigating the risk of information loss.
>
> *Empirical Comparison with Chunk-Based Retrieval Methods*: To empirically validate the efficacy of RAPTOR against traditional chunk-based retrieval methods, we conducted new experiments comparing RAPTOR with SBERT, BM25, and DPR retrievers on the QuALITY, NarrativeQA, and QASPER datasets, focusing solely on the original, chunked documents. RAPTOR consistently outperforms across all datasets and retrievers, highlighting the advantage of our hierarchical, summarization approach over standard chunk concatenation methods. We show the results below:
>
> #### QuALITY
>
> | Model             | Accuracy |
> |---------------------------|----------|
> | **SBERT with RAPTOR**     | **56.6%**    |
> | SBERT without RAPTOR    | 54.9%    |
> | **BM25 with RAPTOR**      | **52.1%**    |
> | BM25 without RAPTOR      | 49.9%    |
> | **DPR with RAPTOR**       | **54.7%**    |
> | DPR without RAPTOR      | 53.1%    |
>
> #### NarrativeQA
>
> | Model                    | ROUGE  | BLEU-1 | BLEU-4 | METEOR |
> |----------------------------------|--------|--------|--------|--------|
> | **SBERT with RAPTOR**            | **30.87%** | **23.50%** | **6.42%**  | **19.20%** |
> | SBERT without RAPTOR               | 29.26% | 22.56% | 5.95%  | 18.15% |
> | **BM25 with RAPTOR**             | **27.93%** | **21.17%** | **5.70%**  | **17.03%** |
> | BM25 without RAPTOR             | 23.52% | 17.73% | 4.65%  | 13.98% |
> | **DPR with RAPTOR**              | **30.94%** | **23.51%** | **6.45%**  | **19.05%** |
> | DPR without RAPTOR                | 29.56% | 22.84% | 6.12%  | 18.44% |
>
> #### Qasper
>
> | Model                | Answer F1 |
> |------------------------------|-----------|
> | **SBERT with RAPTOR**        | **36.70%**    |
> | SBERT without RAPTOR        | 36.23%    |
> | **BM25 with RAPTOR**         | **27.00%**    |
> | BM25 without RAPTOR           | 26.47%    |
> | **DPR with RAPTOR**          | **32.23%**    |
> | DPR without RAPTOR           | 31.70%    |
>
> *Layer-wise Retrieval Analysis Demonstrating RAPTOR's Adaptability*: We also present a layer-wise analysis of the nodes retrieved by RAPTOR across different datasets. This analysis reveals that RAPTOR adaptively utilizes different tree layers (summary vs. leaf nodes) based on the question type and dataset characteristics. For instance, in NarrativeQA, which tends to have more thematic questions, RAPTOR leverages a higher proportion of summary nodes. Conversely, in QASPER, where questions are more extractive, there is a greater reliance on leaf nodes. This adaptability highlights RAPTOR's capability to effectively balance both low-level details and higher-level summaries. We present all of the results below.
>
> ####  % of nodes from different layer with DPR as the retriever
> | Layer | NarrativeQA | Quality | Qasper |
> |-------|-------------|---------|--------|
> | 0     | 42.64       | 67.71   | 77.07  |
> | 1     | 45.00       | 29.43   | 21.88  |
> | 2     | 10.57       | 2.85    | 1.05   |
> | 3     | 1.78        | -       | -      |
> | 4     | 0.003       | -       | -      |
>
> #### % of nodes from different layer with SBERT as the retriever
> | Layer | NarrativeQA | Quality | Qasper |
> |-------|-------------|---------|--------|
> | 0     | 63.22       | 75.59   | 81.51  |
> | 1     | 31.51       | 22.78   | 17.84  |
> | 2     | 4.85        | 1.63    | 0.65   |
> | 3     | 0.42        | -       | -      |
>
> #### % of nodes from different layer with BM25 as the retriever
> | Layer | NarrativeQA | Quality | Qasper |
> |-------|-------------|---------|--------|
> | 0     | 65.04       | 67.64   | 77.24  |
> | 1     | 28.79       | 28.85   | 21.57  |
> | 2     | 5.36        | 3.51    | 1.19   |
> | 3     | 0.81        | -       | -      |
>
> We have added all of these new experimental results to the revised paper.

---

> > ### Author Response · Authors · 2023-11-22
> > **Response to Reviewer YAgR [2/3]**
> >
> > > **Even if clustering is automatically done, how to segment texts into chunks is still left as a problem.**
> >
> > We would like to thank the reviewer for bringing this to our attention. To address this concern, we have expanded the description in our revised manuscript and provide further details below to describe how we perform text segmentation in RAPTOR.
> >
> > In our methodology, we segment texts into chunks of 100 tokens. We ensure that a sentence is not split across different chunks. If a sentence exceeds the 100-token limit, we move the entire sentence to the next chunk, rather than cutting it mid-sentence. This preserves the contextual and semantic coherence of the text within each chunk.
> >
> > Although we perform text segmentation as described above, RAPTOR can be used with any text segmentation method. We believe this flexibility is in fact a strength of the method and offers opportunities for further performance improvement via better text segmentation.
> >
> > > **When comparing model performances, the parameter sizes of models should be the same or similar. However, the paper compares models with totally different parameters.**
> >
> > We agree that it is crucial for a fair assessment to compare models under the same conditions, namely matching parameter counts. To address this concern, in our controlled experiments, we have ensured consistency and fairness by using the same language models across different retrieval methods. Specifically, when evaluating RAPTOR, DPR, and BM25, we employed identical language models - namely, the GPT models and UnifiedQA - for all retrieval methods. This approach allows us to disentangle the impact of the retrieval mechanism from the inherent capabilities of the language models themselves.
> >
> > By maintaining this consistency in language model selection, our experiments accurately reflect the performance differences attributable to the retrieval strategies, rather than differences in language model parameter counts.
> >
> > > **How many retrieved instances are used for baseline models needs to be described in detail. Thus, judging the fairness of the comparison in the experiment is difficult.**
> >
> > We agree that it is important to describe the number of retrieved instances for the baseline models, so we have provided these details below and in our revised manuscript. To ensure a fair comparison, we adopted a token-based approach rather than a direct top-k retrieved instances metric.
> >
> > In our experiments, both RAPTOR and the baseline methods (DPR and BM25) were provided with an equivalent amount of context. Specifically, when utilizing GPT-based models, we allocated 2000 tokens of context to both RAPTOR and the baselines. In the case of the UnifiedQA model, this context was limited to 400 tokens.
> >
> > This token-based approach is crucial for maintaining fairness. We see that the average token count in RAPTOR's summary nodes is typically higher than in the leaf nodes. Directly comparing top-k retrieved instances could inadvertently give RAPTOR an advantage by giving it access to more information. By standardizing the context length in terms of token count, we mitigate this potential bias. In practice, for the baselines, a 2000-token context corresponds approximately to the top-20 retrieved instances, while a 400-token context aligns with around the top-4 retrieved instances.
> >
> > We have updated the appendix of our paper to include these details so that readers can better judge the fairness of our comparisons.

---

> ### Author Response · Authors · 2023-11-22
> **Response to Reviewer YAgR [3/3]**
>
> ### Questions
> > **How did you segment texts into chunks? If you use obvious segments in text like paragraphs, please state it clearly.**
>
> Please see our response above on text segmentation.
>
> > **How did you generate summaries by GPT-3.5-turbo? You should show the actually used prompt.**
>
> Thanks for pointing this out! Our prompt to generate summaries by GPT-3.5-turbo is: Write a summary of the following, including as many key details as possible: {context}. We have added this information to a new section entitled “Summarization Prompt” of the Appendix.
>
> > **What kind of embedding did you use for retrieving chunks and clusters? Did you use the centroids for each cluster or embedding of the summaries? Also, are these calculated by SBERT similar to the clustering step?**
>
> We used SBERT embeddings for retrieving chunks and clusters. We used embeddings of the summaries, which were calculated by SBERT as in the clustering step. We updated the paper to make this more clear. Thanks for pointing out that this wasn’t clear in the original.
>
> > **This is related to the above weakness part. How many retrieved instances are used for baseline models?**
>
> Please see our response above on the number of retrieved instances.
>
>
> > **In the part, "Comparison to State-of-the-art Systems", you compared your RAPTOR with GPT-4 to LongT5 that has 3B of parameters and its variant, ColtT5. Considering that the detailed parameter size of GPT-4 is not released and the size of the baselines' parameters, this comparison is not fair. Did you adopt RAPTOR to LongT5 or ColtT5?**
>
> In our study, the primary objective of the controlled experiments was to evaluate the efficacy of RAPTOR as a retrieval method. To ensure a fair and consistent comparison, we used the same language models (GPT models and UnifiedQA) across different retrieval strategies. This approach allows us to directly assess the impact of the retrieval mechanism on performance.
>
> Regarding the comparison with LongT5 and ColtT5, we note that the state-of-the-art results reported for these models on the datasets in question were achieved using a non-retrieval-based approach, where the entire document is provided for question-answering. This is fundamentally different from our retrieval-augmented setup. Moreover, the model weights for ColtT5 are not publicly available, and only the base LongT5 model weights are accessible. The state-of-the-art scores for LongT5 were derived from versions of the model specifically fine-tuned for QA tasks.
>
> In our case, we employed the UnifiedQA model, a T5-based model with 3B parameters, but crucially, with a context length limit of 512 tokens, which is in line with our retrieval-focused evaluation framework.
>
> > **Similar to the previous question, in Table 5, you should have adopted RAPTOR to the model with almost the same parameter size as that of baselines.**
>
> Please see our response above in regards to model parameter counts.

---

> > ### Author Response · Authors · 2023-11-23
> >
> > Thank you once again for your valuable feedback! As we approach the end of our discussion period, we wanted to check in to see if our previous response has addressed your concerns and answered your questions. If you have any additional questions or if there are any concerns that we haven't addressed, please let us know and we would be happy to answer them.

---

> > > ### Comment · Reviewer_YAgR · 2023-11-23
> > > **You have cleared almost all my concerns**
> > >
> > > Thank you for answering my questions. I appreciate your effort to polish up your paper. Except for the concern in the part below, you have cleared my concerns.
> > > > When comparing model performances, the parameter sizes of models should be the same or similar. However, the paper compares models with totally different parameters. This is problematic from the viewpoint of fairness.
> > >
> > > More specifically, using T5 in your approach is the best configuration to check the performance improvement in models with limited parameters. However, attempting this in the limited rebuttal period is difficult, and the effectiveness of your approach as a retriever has already been guaranteed in the current setting, Considering them, I'll update my score.

---

### Official Review · Reviewer_VSBj · 2023-11-03

**Soundness:** 2 fair
**Presentation:** 2 fair
**Contribution:** 2 fair
**Rating:** 6
**Confidence:** 4

**Summary:**

The authors present an approach to generate recursive summaries for closed-book QA tasks such as NarrativeQA, Qasper, and Quality. The summaries compress multiple documents, which are clustered according to GMM. Then, at test time, the summaries and original passages are retrieved from to perform the task in a RAG-like fashion.

**Strengths:**

1. Strong results compared to multiple baselines, but some choice of baselines are poorly justified and claims of SOTA are not correct.

2. An interesting approach to generate a set of summaries to retrieve from. The nature of the summaries is probably the main value here, especially given that the tree structure of the summaries are mostly ignored. Further, despite other complains for easy scalability, probably this approach will not immediately scale for larger retrieval datasets. Perhaps the focus should be for closed book QA.

3. Results reported on three datasets of varying domains using both closed and open source LMs.

**Weaknesses:**

1. There is very little analysis of the summarized outputs. The analysis included in the main text, Table 6, is difficult to understand and does not reveal much about the content of the summaries. Based on an in-line example, it seems the benefit of this approach may be the abstractive nature of the summaries, and it would be helpful to verify further whether this is the case or some other property is helping.

2a. The baselines in table 1 and 2 are not well justified. It seems like BM25 and DPR do not use the tree structure, also, SBERT can be applied without tree structure.

2b. There are multiple claims of state of the art that are not accurate. In zeroscrolls, flan-ul2 is reported to get 56.9F1 on Qasper. GPT-4 gets 89.2 on Quality in the same paper.

3. The paper is hard to read at times. Details about beam search are confusing (e.g. do we return all the visited docs, or a single sequence of them?).

**Questions:**

Q1: Did you consider using a hierarchical clustering algorithm like agglomerative clustering?

Q2: Why use GMM? In practice, do you assign passages/summaries to more than one cluster?

Q3: Is 4 percent hallucination considered standard? What amount of hallucination have other LLM-based summarization systems experienced?

Q4: Is the example for hallucination in the appendix the full length? I am surprised to see the summary and child node are almost the same size.

Q5: Is the cumulative cosine score used to select the next top-k nodes in beam search approach? If not, then perhaps it makes sense to rename this---the approach does not seem very beam search like.

Q6: When using beam search, will there always be k * d nodes kept? Or is it only d nodes, where you choose the sequence of nodes with the highest cumulative score?

Q7: Why did you need to create your own story? Can you share more details about how the 2600 word story is created?

Q8: Can you provide more details about the difference between collapsed and beam in Figure 3? Does collapsed tend to choose more summaries than beam? Why is top-3 not centered over the 1000 x-ticker? How should we interpret the context length x-axis wrt beam?

Presentation

When referencing the appendix, perhaps specify where in the appendix (i.e. which appendix section).

It is deceptive and not clear to describe RAPTOR as scaling linearly. The actual complexity is closer to O(n * k) where N is the number of tokens and k is the number of layers. It only appears linear because you are using a small k. I suggest to refine the writing to mention it "scales linearly in practice" or something along those lines.

Can make it more clear whether using dev or test data. Also, if using dev data to claim state-of-the-art, then perhaps provide some clarification why test is not being used.

Related Work

Sun et al 2021. Do Long-Range Language Models Actually Use Long-Range Context?---This work pre-dates Liu et al and effectively shows transformers do not handle large contexts effectively.

Since your approach is meant to make up for shortcomings of chunked passages, then you may be interested in "decontextualization". For example, see: https://arxiv.org/abs/2305.14772

Gao et al 2023. Enabling Large Language Models to Generate Text with Citations---This recent work uses summaries in retrieval augmented generation.

Min et al 2021. Joint Passage Ranking for Diverse Multi-Answer Retrieval---This work does tree-based retrieval for open-domain QA.

---

> ### Author Response · Authors · 2023-11-22
> **Response to Reviewer VSBj [1/4]**
>
> Thank you for the time and care you put into this review! We are pleased that you found our work interesting and we have used your feedback to greatly improve the paper, running new experiments, providing new analyses and figures, and updating the text for clarity.
>
> Weakness 1:
> > **There is very little analysis of the summarized outputs. The analysis included in the main text, Table 6, is difficult to understand and does not reveal much about the content of the summaries. Based on an in-line example, it seems the benefit of this approach may be the abstractive nature of the summaries, and it would be helpful to verify further whether this is the case or some other property is helping.**
>
> The reviewer raises a good point. We also believe that the abstractive nature of the summaries is beneficial, so we ran new ablation studies to more clearly demonstrate the impact of abstractive summarization.
>
> With the collapsed tree method (the best performing variant of our method), tree structure has no effect at retrieval time, but it may influence the abstractive summaries formed at build time. In response to weakness 2, we show that RAPTOR, controlled for the retriever, outperforms the leaf node only baseline, highlighting the benefit of abstractive summaries. We provide a subset of the results here:
>
> #### Table 1: QuALITY
> | Model  | Accuracy |
> |---------------------------|----------|
> | **SBERT with RAPTOR**     | **56.6%**    |
> | SBERT without RAPTOR    | 54.9%    |
>
> #### Table 2: NarrativeQA
> | Model  | ROUGE  | BLEU-1 | BLEU-4 | METEOR |
> |----------------------------------|--------|--------|--------|--------|
> | **SBERT with RAPTOR**            | **30.87%** | **23.50%** | **6.42%**  | **19.20%** |
> | SBERT without RAPTOR               | 29.26% | 22.56% | 5.95%  | 18.15% |
>
> #### Table 3: Qasper
> | Model | Answer F1 |
> |------------------------------|-----------|
> | **SBERT with RAPTOR**        | **36.70%**    |
> | SBERT without RAPTOR        | 36.23%     |
>
>
>
> We further conduct an ablation study across all three datasets and across three different retrievers with RAPTOR to see what layers the retrieved nodes come from. We observe that from 18.5% to 57% of the retrieved nodes are non-leaf nodes (abstractive summaries).
>
> #### % of nodes from non-leaf nodes
> | Dataset | DPR    | SBERT  | BM25   |
> |------------|--------|--------|--------|
> | NarrativeQA| 57.36 | 36.78 | 34.96 |
> | Quality    | 32.28| 24.41 | 32.36 |
> | Qasper     | 22.93 | 18.49 | 22.76 |
>
> ####  % of nodes from different layer with DPR as the retriever
> | Layer | NarrativeQA | Quality | Qasper |
> |-------|-------------|---------|--------|
> | 0     | 42.64       | 67.71   | 77.07  |
> | 1     | 45.00       | 29.43   | 21.88  |
> | 2     | 10.57       | 2.85    | 1.05   |
> | 3     | 1.78        | -       | -      |
> | 4     | 0.003       | -       | -      |
>
> #### % of nodes from different layer with SBERT as the retriever
> | Layer | NarrativeQA | Quality | Qasper |
> |-------|-------------|---------|--------|
> | 0     | 63.22       | 75.59   | 81.51  |
> | 1     | 31.51       | 22.78   | 17.84  |
> | 2     | 4.85        | 1.63    | 0.65   |
> | 3     | 0.42        | -       | -      |
>
> #### % of nodes from different layer with BM25 as the retriever
> | Layer | NarrativeQA | Quality | Qasper |
> |-------|-------------|---------|--------|
> | 0     | 65.04       | 67.64   | 77.24  |
> | 1     | 28.79       | 28.85   | 21.57  |
> | 2     | 5.36        | 3.51    | 1.19   |
> | 3     | 0.81        | -       | -      |
>
> Since the reviewer found Table 6 difficult to understand, we updated the paper to better describe it. Columns represent different starting points (highest layer) and rows represent different numbers of layers queried from that starting layer. For example, the second row (2 layers) and third column (Layer 2) corresponds to retrieval from the second and first layers, excluding the leaf nodes (0th layer). Thanks for pointing out that this was unclear!
>
> In addition to the above, we also conducted a new ablation study using a recency-based tree approach, where we recursively encoded and summarized contiguous chunks of text, setting the window size to the average RAPTOR cluster size (~ 6.7 nodes, thus a window size of 7 nodes). The results of this ablation are as follows:
>
> | Configuration | Accuracy |
> |--------------------------------------|------------|
> | **RAPTOR + SBERT embeddings + UnifiedQA** | **56.6%**      |
> | Recency-based tree + SBERT embeddings + UnifiedQA | 55.8% |
>
> This shows that the clustering mechanism in RAPTOR provides an improvement in accuracy over a recency-based tree approach and that abstractive summaries due to the RAPTOR’s clustering algorithm (and resulting tree structure) help as opposed to just any abstractive summarization. We have added this new ablation study to the paper.
>
> To enable external researchers to perform further analysis of the summarized outputs, we will also release all RAPTOR summaries/trees as binaries.

---

> ### Author Response · Authors · 2023-11-22
> **Response to Reviewer VSBj [2/4]**
>
> Weakness 2:
> > **The baselines in table 1 and 2 are not well justified. It seems like BM25 and DPR do not use the tree structure, also, SBERT can be applied without tree structure.**
>
> We thank the reviewer for the suggestion of testing each retriever with and without RAPTOR to form a clearer set of baselines. We ran this experiment using SBERT, BM25, and DPR with and without the RAPTOR tree structure on all three datasets: QASPER, NarrativeQA and QuALITY with the UnifiedQA model and add these results to the paper under “Further Experimental Results” (and show below). RAPTOR with any retriever consistently outperforms the respective retriever (without RAPTOR) across all datasets.
>
> #### Table 1: QuALITY
>
> | Model             | Accuracy |
> |---------------------------|----------|
> | **SBERT with RAPTOR**     | **56.6%**    |
> | SBERT without RAPTOR    | 54.9%    |
> | **BM25 with RAPTOR**      | **52.1%**    |
> | BM25 without RAPTOR      | 49.9%    |
> | **DPR with RAPTOR**       | **54.7%**    |
> | DPR without RAPTOR      | 53.1%    |
>
> #### Table 2: NarrativeQA
>
> | Model                    | ROUGE  | BLEU-1 | BLEU-4 | METEOR |
> |----------------------------------|--------|--------|--------|--------|
> | **SBERT with RAPTOR**            | **30.87%** | **23.50%** | **6.42%**  | **19.20%** |
> | SBERT without RAPTOR               | 29.26% | 22.56% | 5.95%  | 18.15% |
> | **BM25 with RAPTOR**             | **27.93%** | **21.17%** | **5.70%**  | **17.03%** |
> | BM25 without RAPTOR             | 23.52% | 17.73% | 4.65%  | 13.98% |
> | **DPR with RAPTOR**              | **30.94%** | **23.51%** | **6.45%**  | **19.05%** |
> | DPR without RAPTOR                | 29.56% | 22.84% | 6.12%  | 18.44% |
>
> #### Table 3: Qasper
>
> | Model                | Answer F1 |
> |------------------------------|-----------|
> | **SBERT with RAPTOR**        | **36.70%**    |
> | SBERT without RAPTOR        | 36.23%    |
> | **BM25 with RAPTOR**         | **27.00%**    |
> | BM25 without RAPTOR           | 26.47%    |
> | **DPR with RAPTOR**          | **32.23%**    |
> | DPR without RAPTOR           | 31.70%    |
>
> Note that we run the above DPR experiments with dpr-multiset-base as opposed to dpr-single-nq-base, because dpr-multiset-base performs better than dpr-single-nq-base in the original DPR paper.
>
> Weakness 3:
> > **The paper is hard to read at times. Details about beam search are confusing (e.g. do we return all the visited docs, or a single sequence of them?)**
>
> Thank you for sharing this feedback. We agree that the details of beam search were unclear, so we renamed the method to reduce confusion (now called “tree traversal”), updated Figure 2, and improved the description in the paper and in the caption of Figure 2, both of which are included below for reference:
>
> Querying Section:
> “Tree traversal first selects the top-k most relevant root nodes based on their cosine similarity to the query embedding. The children of these selected nodes are considered at the next layer and the top-k nodes are selected from this pool, again based on their cosine similarity to the query vector. This process is repeated until the leaf nodes are reached. Finally, the text from all selected nodes is concatenated to form the retrieved context.”
>
> Caption of Figure 2:
> “Tree traversal starts at the root level of the tree and retrieves the top-k (here, top-1) node(s) based on cosine similarity to the query vector. At each level, it retrieves the top-k node(s) from the child nodes of the previous layer’s top-k. Collapsed tree collapses the tree into a single layer and retrieves nodes until a threshold number of tokens is reached, based on cosine similarity to the query vector. The nodes on which cosine similarity search is performed are highlighted in both illustrations.”
>
> Weakness 4:
> > **There are multiple claims of state of the art that are not accurate. In zeroscrolls, flan-ul2 is reported to get 56.9F1 on Qasper. GPT-4 gets 89.2 on Quality in the same paper.**
>
> In our paper, we evaluate each method on the full test set for QuALITY, QASPER, and NarrativeQA; in contrast, the zeroscrolls paper evaluates on only a subset of 500 examples for each dataset, meaning that their results are not directly comparable.
>
> Further, our SOTA claims for QuALITY are based on the QuALITY leaderboard, where we submitted our results on the hidden test set: https://nyu-mll.github.io/quality/.

---

> > ### Comment · Reviewer_VSBj · 2023-11-22
> >
> > Thank you for the new results. RAPTOR provides a relatively small improvement on QUALITY compared to what's seen in the paper (table 5) where RAPTOR outperforms the next best model by more than 20 accuracy. Can you help me understand why this would be the case?
> >
> > Overall I am grateful for the effort in the rebuttal, but i wanted to address this point quickly since it immediately stood out.

---

> > > ### Author Response · Authors · 2023-11-22
> > >
> > > We are glad that the reviewer appreciated the effort we put into this rebuttal, and we thank them for this follow-up question regarding the performance improvement observed with RAPTOR on the QuALITY dataset. We would like to clarify that the experiments detailed in our rebuttal were conducted using the UnifiedQA model, which is a 3B parameter model with a maximum input context length of 512 tokens. For these experiments, we provided the model with 400 tokens of context, allowing for an average of around 3 RAPTOR summary nodes and about 4 leaf nodes, given the average token counts of 131 and 85, respectively.
> > >
> > > In contrast, the result reported in Table 5 of our paper was achieved using a more powerful model, namely GPT-4, to which we supplied 2000 tokens of context. This larger model is better able to use the context provided by RAPTOR and had access to a large number of RAPTOR summary nodes, enhancing its performance on the task. The choice to conduct the majority of our rebuttal experiments with an open-source model, such as UnifiedQA, instead of GPT-4, was driven by the goal of cost-effectiveness. Running extensive experiments with GPT-4 would have been prohibitively expensive and more difficult for other researchers to reproduce.

---

> > > > ### Comment · Reviewer_VSBj · 2023-11-22
> > > >
> > > > So would you say the result in the paper has a big gap solely because of GPT-4? Does RAPTOR only give a small bonus over simpler retrieval w GPT-4?

---

> > > > > ### Author Response · Authors · 2023-11-22
> > > > >
> > > > > We see clear benefits from RAPTOR across all retrievers, models, and tasks on which we tested, as demonstrated in the controlled experiments presented. The best variant we tried used GPT-4 and collapsed tree RAPTOR retrieval, leading to 82.6 on QuALITY. We have not tried simpler retrieval with GPT-4 due to cost constraints, but we of course acknowledge that the strength of GPT-4 contributed to that result.
> > > > >
> > > > >
> > > > > The degree of improvement from RAPTOR does seem to be model-dependent, as demonstrated in our experiments with the Llama2 7B chat model on the QASPER dataset, where RAPTOR's integration led to a more than 5% increase in performance over the baseline retrieval method.
> > > > >
> > > > >
> > > > > Thank you for this follow-up question and we hope that you found the rest of our rebuttal helpful as well.

---

> > > > > > ### Comment · Reviewer_VSBj · 2023-11-22
> > > > > >
> > > > > > I've increased my score because although I think RAPTOR has flaws, it is clever and should inspire future work. The new results help.
> > > > > >
> > > > > > But I think you should make the writing more clear that GPT-4 is doing a lot of the heavy lifting compared to those baselines. Perhaps should add context length to the table 5 or mention it. This does not seem right "For all other experiments, we provide 2000 tokens of context to RAPTOR and the baselines."

---

> > > > > > > ### Author Response · Authors · 2023-11-23
> > > > > > >
> > > > > > > Thank you for considering our rebuttal and new experimental results! We are happy to clarify the paper and appreciate the reviewer’s suggestions on how to do so.

---

> ### Author Response · Authors · 2023-11-22
> **Response to Reviewer VSBj [3/4]**
>
> ### Questions
>
> > **Did you consider using a hierarchical clustering algorithm like agglomerative clustering? Why use GMM? In practice, do you assign passages/summaries to more than one cluster?**
>
> We did not experiment with hierarchical clustering algorithms, such as agglomerative clustering. These algorithms have a time complexity of O(N^2 log N), which may be prohibitively expensive. Furthermore, these algorithms assign each data point to a single cluster. Our goal in RAPTOR is to create higher-level representations of text, which means the semantics of a chunk should be able to be represented across different nodes in our tree representation.
>
> We selected GMM for RAPTOR's clustering, because the target task necessitated both automated cluster determination and the capability for nodes to belong to multiple clusters.
>
> Empirically, we saw that soft clustering occurs when the length of the input document is greater than 14,300 tokens. Across all datasets, 74.3% of the documents with an input length greater than 14,300 tokens contain some amount of soft clustering. Amongst the documents with soft clustering, 38.1% of the summary nodes contain soft clustered child nodes.
>
> We acknowledge that better clustering methods could further improve the performance of RAPTOR, and welcome future work in this area.
>
> > **Is 4 percent hallucination considered standard? What amount of hallucination have other LLM-based summarization systems experienced?**
>
> Xu et al. (2023) [1] conducted a faithfulness analysis on summaries generated by GPT 3.5, the same model we employed for our summarization tasks, checking if the summary is entailed by the retrieved documents. They found that the non-faithfulness (i.e. hallucination) rates in summaries ranged from 3% to 26% across different datasets. Our observed hallucination rate of 4% falls on the lower end of this spectrum.
>
> [1] [Xu et al., 2023. RECOMP: Improving Retrieval-Augmented LMs with Compression and Selective Augmentation](https://arxiv.org/pdf/2310.04408.pdf)
>
> > **Is the example for hallucination in the appendix the full length? I am surprised to see the summary and child node are almost the same size.**
>
> The example for hallucination in the appendix is not full length. We truncated both the child and parent node to illustrate the types of hallucinations that can occur. To satisfy your curiosity, we provide an analysis on the lengths of the summary and child nodes below.
>
> The average ratio of the summary length to the sum of child node lengths across all datasets is 0.28 (i.e. a 72% compression rate). Since the parent node summarizes multiple nodes, it is possible, though unlikely, for the summary node and a single child node to be the same length. Across all datasets, the average summary length is 131 tokens while the average child node length is 86 tokens. We provide statistics on all three datasets below:
>
> | Dataset        | Avg. Summary Length (tokens) | Avg. Child Node Text Length (tokens) | Avg. # of Child Nodes Per Parent | Avg. Compression Ratio (Summary to Cluster Text Tokens) (%) |
> |----------------|------------------------------|--------------------------------------|----------------------------------|------------------------------------------------------------|
> | All Datasets   | 131   | 85.6    | 6.7                | .28                                                       |
> | Quality        | 124.4   | 87.9        | 5.7               | .28                                                      |
> | NarrativeQA    | 129.7      | 85.5           | 6.8           | .27                                                       |
> | Qasper         | 145.9          | 86.2          | 5.7              | .35                                                       |
>
> We have added these statistics to the appendix of the paper.
>
> > **Is the cumulative cosine score used to select the next top-k nodes in beam search approach? If not, then perhaps it makes sense to rename this---the approach does not seem very beam search like. When using beam search, will there always be k * d nodes kept? Or is it only d nodes, where you choose the sequence of nodes with the highest cumulative score?**
>
> Thank you for pointing this out! We have renamed the beam search approach to “tree traversal” to better reflect the nature of this method.
>
> Beam search (now renamed tree traversal) retrieves text from k * d nodes. In practice, for the majority of our experiments (and unless otherwise noted), we use the collapsed tree method. We have updated the description of tree traversal in the paper for clarity.
>
> > **Why did you need to create your own story? Can you share more details about how the 2600 word story is created?**
>
> We created our own story to ensure that we were testing the models on data on which they had not been trained. We have added an excerpt in the appendix, as well as a link to the full story.

---

> ### Author Response · Authors · 2023-11-22
> **Response to Reviewer VSBj [4/4]**
>
> > **Can you provide more details about the difference between collapsed and beam in Figure 3? Does collapsed tend to choose more summaries than beam? Why is top-3 not centered over the 1000 x-ticker? How should we interpret the context length x-axis wrt beam?**
>
> Figure 3 presents a comparison between two querying methods—collapsed tree and beam search—across a range of context lengths on the QASPER dataset. The top-k value tickers correspond to the average number of tokens within the respective top-k groups. The top-3 label is not perfectly centered over the 1000 mark on the x-axis, because the average context length for the top-3 category doesn't align exactly with 1000 tokens.
>
> The beam search method (now renamed tree traversal) will always use a consistent ratio of information from the leaf nodes, specifically 1 / {total number of layers} in the tree. On the other hand, as shown in the table below, the collapsed tree approach offers greater flexibility in the ratio of information from the summary nodes to the leaf nodes. Furthermore, for NarrativeQA, which includes more thematic questions, around 47% of nodes are summary nodes, whereas in QASPER, which consists of more extractive questions than abstractive questions, collapsed tree retrieves 81% leaf nodes and only 19% summary nodes.
>
> #### % of nodes from different layer with SBERT as the retriever
> | Layer | NarrativeQA | Quality | Qasper |
> |-------|-------------|---------|--------|
> | 0     | 63.22       | 75.59   | 81.51  |
> | 1     | 31.51       | 22.78   | 17.84  |
> | 2     | 4.85        | 1.63    | 0.65   |
> | 3     | 0.42        | -       | -      |
>
> We have added this new analysis to the appendix of the revised paper.
>
>
> > **When referencing the appendix, perhaps specify where in the appendix (i.e. which appendix section).**
>
> Thank you for this suggestion! We have updated the paper accordingly.
>
> > **It is deceptive and not clear to describe RAPTOR as scaling linearly. The actual complexity is closer to O(n * k) where N is the number of tokens and k is the number of layers.**
>
> Thank you for your insights regarding the O(n * k) complexity of the RAPTOR algorithm. We acknowledge your point and wish to present an empirical observation. Across all trees the average number of nodes in a parent layer is 18.2% of the number of nodes in the child layer. Similarly, across all datasets, the average token count of the parent layer is 26.1% of the token count of all nodes in the child layer.
>
> This shows that in each layer of RAPTOR, the number of tokens processed decreases by more than half. This reduction significantly impacts the overall complexity. The total computational work across all layers can be described as a geometric series:
>
> - $$ N + \frac{N}{2} + \frac{N}{4} + \frac{N}{8} + ... $$
>
> Summing up this series, the total computational effort is found to be 2N, which simplifies to a linear O(N) complexity.
>
> Thus, RAPTOR’s complexity scales linearly with the number of tokens (and independently of the number of layers), so long as the “compression” factor between layers is at least 50%, which empirically holds across all of our experiments.
>
> > **Can make it more clear whether using dev or test data. Also, if using dev data to claim state-of-the-art, then perhaps provide some clarification why test is not being used.**
>
> All of RAPTOR’s state-of-the-art results are reported on the test data. For QuALITY, we make our SOTA claims based on the QuALITY leaderboard, where we had submitted our results on the hidden test set: https://nyu-mll.github.io/quality/. For controlled experiments against baselines, both RAPTOR and baselines results are reported on the dev dataset only for QuALITY, since the test set answers are hidden.
>
> > **Related Work**
>
> Thank you so much for pointing us to relevant literature! We have updated our paper to include these references.

---

### Author Response · Authors · 2023-11-23

We have really taken on board the need to emphasize comparable experiments where the only difference is using RAPTOR summarization or not. While we believe it is still reasonable to show our best new scores (using GPT-4) relative to previous work on a leaderboard, we will mainly emphasize truly comparable experiments, as in the revised PDF, which includes the new comparable results that we have generated in the rebuttal period.

---

### Meta-Review · Area_Chair_K6mb · 2023-12-12

**Metareview:**

All four reviewers are positive about the paper (two accepts and two borderline accepts). Initially, the reviewers requested additional analysis and raised concerns about misleading claims regarding the results and SOTA performance. The rebuttal was convincing, although some reviewers questioned the fact that, for some experiments in the paper, GPT-4 is doing most of the heavy lifting. Overall, the AC agrees with the reviewers that the idea of the proposed approach is interesting, and recommends acceptance.

**Justification For Why Not Higher Score:**

For some experiments in the paper, the initial claims about significant gains over SOTA were misleading, given that the accuracy gap was mostly due to the use of GPT-4

**Justification For Why Not Lower Score:**

The idea is simple, novel, and sensible, as acknowledged by all reviewers

---

### Decision · Program_Chairs · 2024-01-16

Accept (poster)